# Unraveling a tumor type-specific regulatory core underlying E2F1-mediated epithelial-mesenchymal transition to predict receptor protein signatures

Faiz M. Khan[1], Stephan Marquardt[2], Shailendra K. Gupta[1,3], Susanne Knoll[2], Ulf Schmitz[1,4,5], Alf Spitschak[2], David Engelmann[2], Julio Vera [6], Olaf Wolkenhauer[1,7] & Brigitte M. Pützer[2]

Cancer is a disease of subverted regulatory pathways. In this paper, we reconstruct the regulatory network around E2F, a family of transcription factors whose deregulation has been associated to cancer progression, chemoresistance, invasiveness, and metastasis. We integrate gene expression profiles of cancer cell lines from two E2F1-driven highly aggressive bladder and breast tumors, and use network analysis methods to identify the tumor type-specific core of the network. By combining logic-based network modeling, in vitro experimentation, and gene expression profiles from patient cohorts displaying tumor aggressiveness, we identify and experimentally validate distinctive, tumor type-specific signatures of receptor proteins associated to epithelial–mesenchymal transition in bladder and breast cancer. Our integrative network-based methodology, exemplified in the case of E2F1-induced aggressive tumors, has the potential to support the design of cohort- as well as tumor type-specific treatments and ultimately, to fight metastasis and therapy resistance.

[1] Department of Systems Biology and Bioinformatics, University of Rostock, 18051 Rostock, Germany. [2] Institute of Experimental Gene Therapy and Cancer Research, Rostock University Medical Center, 18057 Rostock, Germany. [3] Department of Bioinformatics, CSIR—Indian Institute of Toxicology Research, Lucknow 206001, India. [4] Gene & Stem Cell Therapy Program, Centenary Institute, Camperdown, NSW 2050, Australia. [5] Sydney Medical School, University of Sydney, Sydney, NSW 2006, Australia. [6] Department of Dermatology, Laboratory of Systems Tumor Immunology, Erlangen University Hospital and FAU University of Erlangen-Nuremberg, 91054 Erlangen, Germany. [7] Stellenbosch Institute for Advanced Study (STIAS), Wallenberg Research Centre at Stellenbosch University, Stellenbosch 7600, South Africa. Faiz M. Khan, Stephan Marquardt, and Shailendra K. Gupta contributed equally to this work. Julio Vera, Olaf Wolkenhauer, and Brigitte M. Pützer jointly supervised this work. Correspondence and requests for materials should be addressed to J.V. (email: Julio.Vera-Gonzalez@uk-erlangen.de)

Recent advances in sequencing and omics technologies provide us with data that can be used to identify and characterize cancer and tumor-specific molecular networks. The analyses of these networks have given insights into various aspects of carcinogenesis, tumor progression, and metastasis[1, 2]. The set of mutated, deregulated, or epigenetically modified cancer genes is highly patient and tumor-type variable, and more important, these genes are integrated in a small set of regulatory pathways[2]. Further, these pathways are not isolated: they crosstalk to shape and fine-tune basic cellular phenotypes that are subverted in cancer. In recent years, several researchers have deployed methodologies based on the reconstruction of cancer-associated networks and used them to analyze high-throughput cancer data[3–5]. Interestingly, it has been found that cancer networks are enriched in regulatory motifs, and beyond, that cancer-related regulatory motifs do crosstalk. Network hubs, feedback, and feedforward loops, the regulatory motifs often encountered in cancer networks, are able to induce a complex regulatory behavior that evades the use of conventional data analysis tools for their understanding[6, 7]. Hence, the utilization of advanced network-based methodologies and mathematical modeling becomes necessary to get a deeper understanding of cancer networks.

An outstanding example of a deregulated cancer network is the one controlled by the E2F family of transcription factors.

The most prominent member of this family, E2F1, is involved in a number of essential cancer-related cellular processes such as proliferation, apoptosis, and differentiation[8]. E2F1 is a remarkable example of a network hub as this protein interacts with many genes, proteins, and other transcription factors through a variety of regulatory mechanisms. In the context of solid tumors, unbalanced E2F1 regulation can lead to the emergence of aggressive tumor cells, which drive cancer progression, resistance to anti-cancer drugs, and the rise of metastatic lesions[9–13].

Enforced E2F1 expression in advanced tumors and metastases of different kinds of cancers correlates with pronounced resistance towards therapy and poor patient prognosis[14, 15]. E2F1 drives epithelial–mesenchymal transition (EMT), similar to the classical EMT inducer TGFB1, via signaling pathways that involve non-coding RNAs[16]. As a direct target of E2F1, enforced expression of *miR-224/452* in melanoma cells stimulates a mesenchymal phenotype by repressing the metastasis suppressor *TXNIP* associated with changes in the actin cytoskeleton towards an enhanced invasive cell behavior. TXNIP in turn controls E2F1 activity in a negative regulatory loop. This process is reversible through ablation of endogenous E2F1 in highly aggressive skin cancer cells, leading to increased TXNIP and E-Cadherin and loss of mesenchymal markers SNAI2, ZEB1, and Vimentin[9, 17].

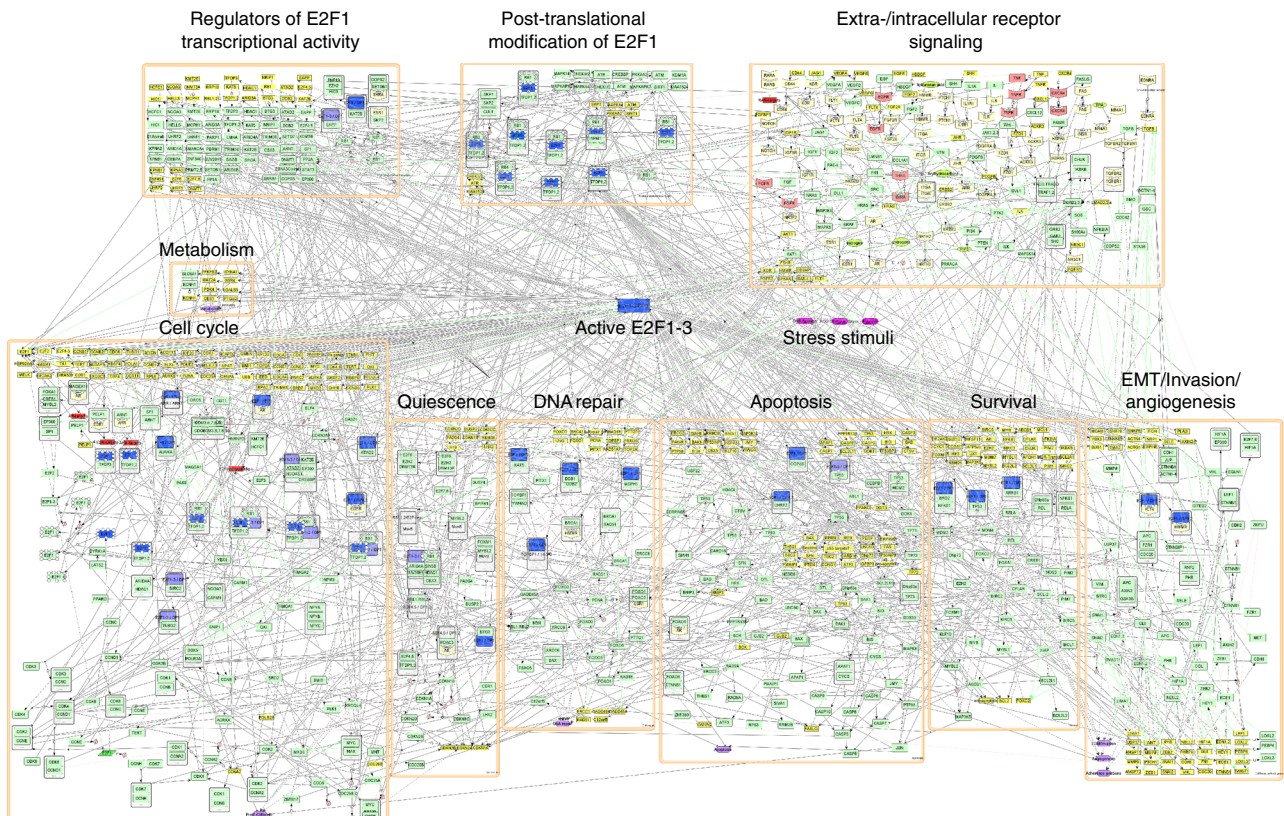

**Fig. 1** A modularized map of E2F1 in tumor progression and metastasis. The map contains three E2F1 regulatory compartments: (i) Extra-/intracellular receptor signaling ($n = 113$); (ii) Post-translational modifications of E2F1 ($n = 24$); (iii) Regulators of E2F1 transcriptional activity ($n = 66$). Furthermore, there are seven functional compartments: (i) Cell cycle ($n = 145$); (ii) Quiescence ($n = 29$); (iii) DNA repair ($n = 33$); (iv) Metabolism ($n = 11$); (v) Apoptosis ($n = 89$); (vi) Survival ($n = 52$); (vii) EMT/invasion/angiogenesis ($n = 69$), where n stands for the number of factors in each compartment. Biomolecules are visualized in standard CellDesigner format (gene: *yellow rectangle*; protein: *light green round cornered box*; receptor: *light yellow hexagon*; ligand: *green oval*; phenotype: *violet hexagon*; drug/external stimulus: *pink box*). For better visualization, in the map transcription and translation are condensed to one reaction directly leading from gene to protein. In *red*, we represent place holders for protein families (e.g., FGFR for FGFR1-4, FGF for FGF1-23, ITGA, and ITGB for alpha and beta integrins) and unspecified genes responding to a given transcription factor. The microRNA layer is not included in this CellDesigner diagram, but in the Cytoscape network provided as Supplementary Information. The interactive E2F1 interaction map can be accessed at: https://navicell.curie.fr/pages/maps_e2f1.html

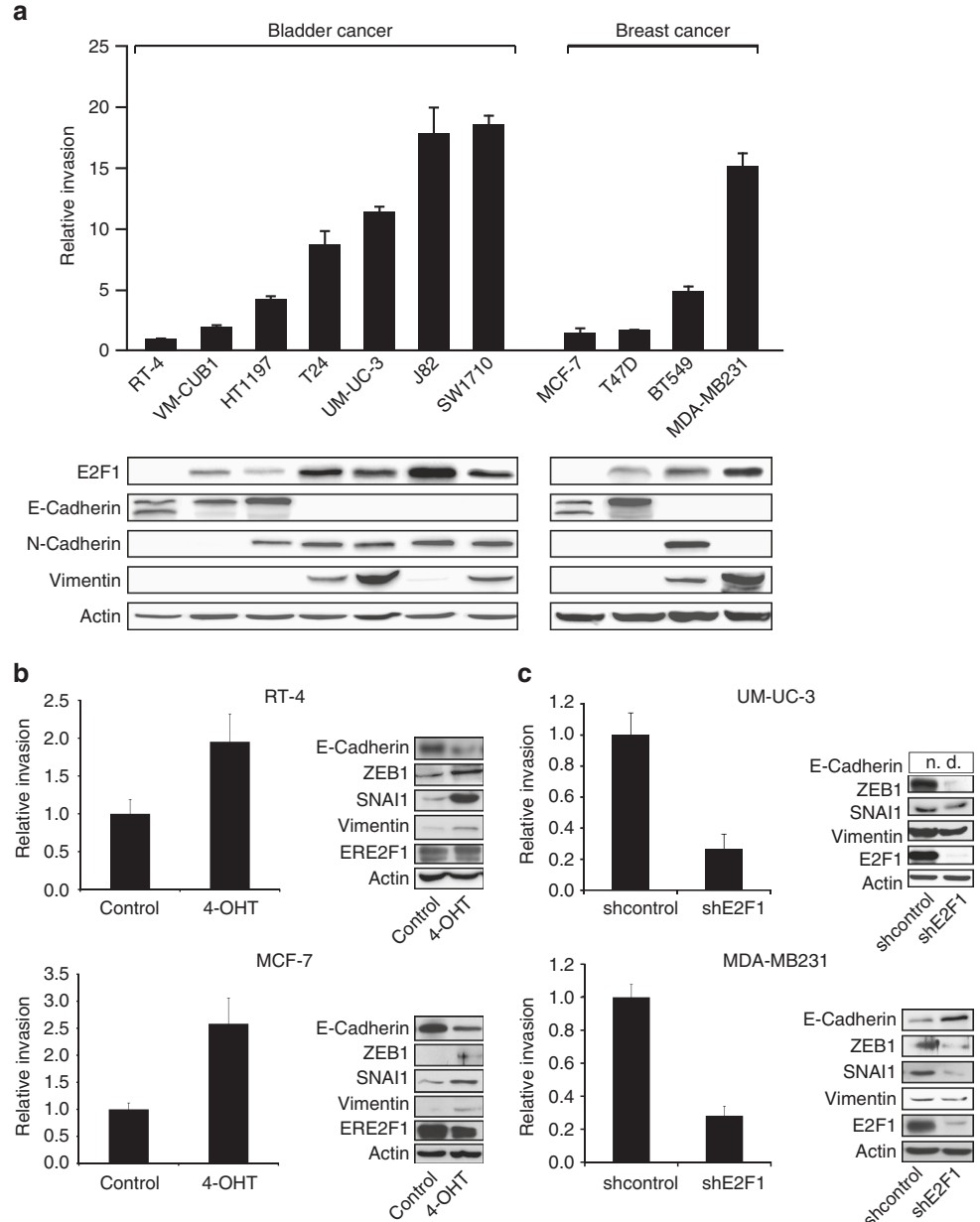

**Fig. 2** Invasive potential and EMT marker expression in less-invasive and invasive human bladder and breast cancer cell lines. **a** Boyden chamber assay and western blot showing the invasive potential and the expression of E2F1 and EMT markers of various bladder and breast cancer lines. RT-4 was used as reference. **b** Indicated cells were transduced with adenoviral vector expressing the 4-OHT responsive estrogen receptor (*ER*)-E2F1 fusion protein (Ad.ER-E2F1) to conditionally activate E2F1 nuclear translocation. After 24 h they were applied to Boyden chamber assay and induced with 4-OHT or ethanol as control (*left panels*) or harvested for protein isolation and western blotting (*right panels*). **c** Cells were transduced with adenoviral vector expressing shE2F1 or shcontrol. After 72 h cell invasion was determined (*left*). Protein level and shE2F1 knockdown is indicated by immunoblots (*right*). All figures are representatives of at least three independent experiments. *Error bars* indicate s.e.m., *n.d.* not detectable

Another mediator of E2F1-induced EMT is *miR-205*[11, 13]. Inhibition of the E-Cadherin repressors *ZEB1* and *ZEB2* by this microRNA results in stabilization of the epithelial cancer cell phenotype[18]. The relevance of this transcription factor to tumor progression was also shown in a genetic model by interbreeding Neu transgenics with *E2F1* knockout mice as well as in *HER2+* breast cancer patients, in which the E2F activation status predicts relapse and metastatic potential of MMTV-Neu-induced tumors[19]. In fact, E2F-responsive genes define a novel molecular subset of high-grade human tumors of the breast, ovary, and prostate, termed ERGO (E2F-responsive gene overexpressing) cancers[20]. Our studies also revealed that vascular endothelial growth factor-C (*VEGF-C*) and its cognate receptor *VEGFR-3*, both highly upregulated in cancer cells with abundant E2F1 expression, are direct targets of this transcription factor[21]. Co-regulation of *VEGF-C*/*VEGFR-3* by E2F1 stimulates endothelial cells to form tubule-like structures and promotes neovascularization in mice. *E2F1* is activated by VEGFR-3 signaling in a positive feedback loop and both proteins cooperate in the nucleus to co-regulate transactivation of the proangiogenic cytokine *PDGF-B*. In addition, we identified the epidermal growth factor receptor (*EGFR*) as a direct target of E2F1 and demonstrated that inhibition of receptor signaling abrogates E2F1-induced invasiveness[9]. These results provide

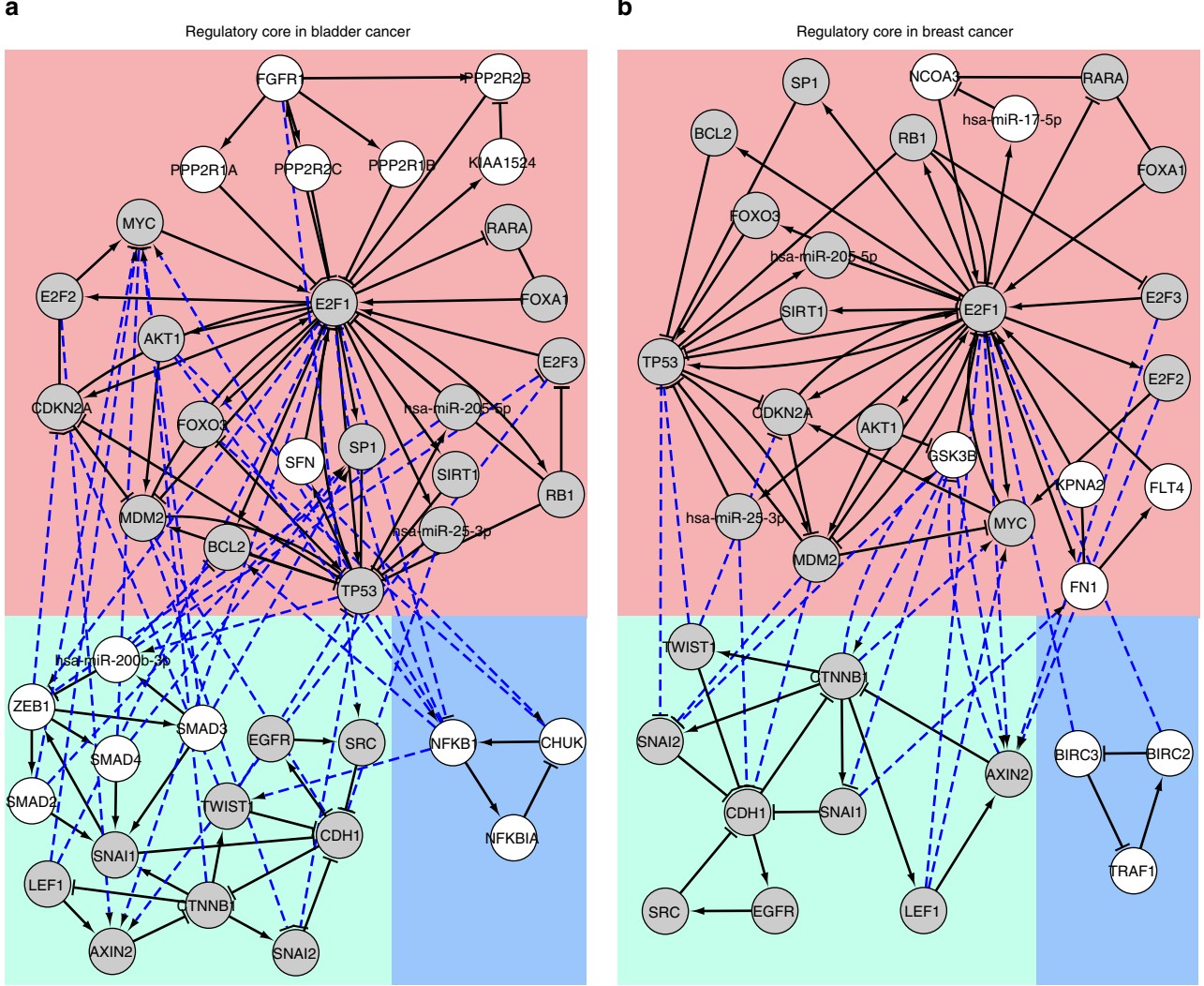

**Fig. 3** Core regulatory networks that drive tumor invasiveness in **a** bladder and **b** breast cancer. The regulatory cores contain three disjoint sub-networks (shown in *blue*, *pink*, and *green* background colors). *Blue dotted lines* represent direct interactions extracted from the comprehensive E2F1 regulatory network to interconnect these sub-networks. The networks were generated using Cytoscape plugin NetDS. *Nodes in gray color* are common in both regulatory core networks, whereas those in *white color* are unique

support for the outstanding, cell context-dependent unique role of E2F1 in driving cancer aggressiveness.

Here, we use a network approach to identify the tumor type-specific regulatory core and to predict receptor protein signatures associated with E2F1-mediated EMT transitions in two types of highly aggressive solid tumors, bladder, and breast cancer. To this end, we construct a comprehensive map of the regulatory network around the E2F family. By mapping gene expression profiles from cancer cell lines displaying the features of EMT transition onto the E2F1 interaction map, we identify a tumor type-specific regulatory core. We then analyze the regulatory core to predict tumor-specific receptor protein signatures linked to aggressiveness. By conducting in vitro experiments, we could verify the impact of these molecules on tumor cell invasiveness. We also found a correlation between the molecular signatures and clinical tumor aggressiveness in relevant patient data.

## Results

**A comprehensive E2F1 interaction map**. To understand how E2F1 interacts with different molecules and how it mediates cancer-related processes, we constructed a functionally modularized interaction map based on information retrieved

from published literature and databases (Fig. 1). The comprehensive map of E2F1 regulation and activity is based on manual exploration of over 800 publications related to E2F1 and other E2F family proteins, as well as connected pathways having a role in cancer-related cellular processes. The map contains 879 nodes including different types of factors (genes, proteins, microRNAs, or complexes) and 2278 interactions. To improve visualization, we modularized the map into several regulatory and functional compartments (Fig. 1). The map comprises HUGO annotations of all the factors, together with meta-information about isoform expression (e.g., DNp73 or mutant TP53) and corresponding PubMed references.

**E2F1 drives EMT in bladder and breast cancer**. Recent clinical results indicate that E2F1 is upregulated in high-grade bladder and breast cancers[14, 22, 23]. To further substantiate these findings, we examined the effects of E2F1 activity on the invasive capacity of patient-derived metastatic bladder and breast tumor cell lines using functional invasion assays, western blotting, and PCR analysis. The experiments revealed a clear correlation of E2F1 expression with the invasive behavior and EMT marker expression in both cell models: high levels of E2F1 and mesenchymal markers in invasive bladder

**a**  Logic-based model of bladder cancer

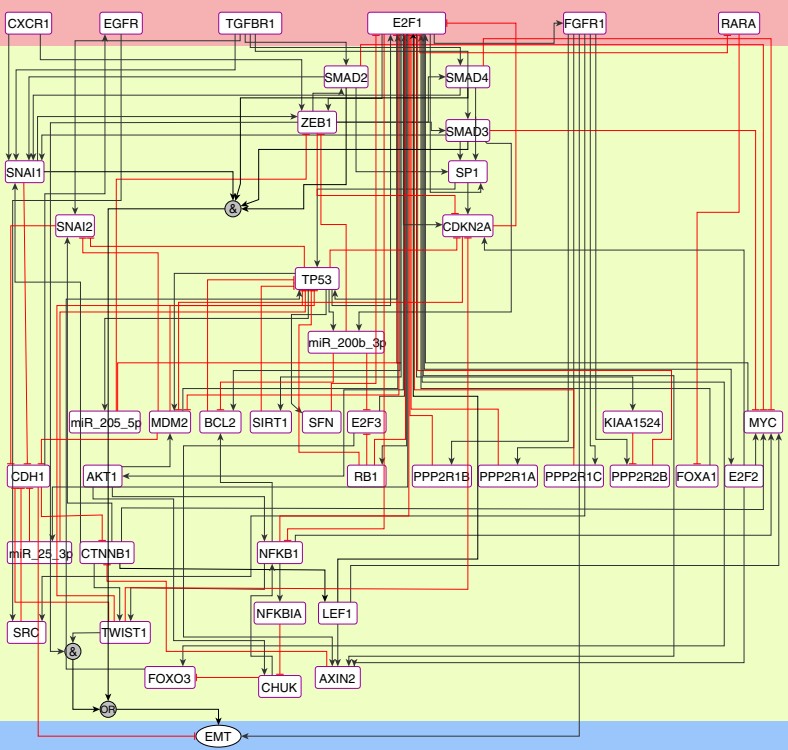

**b**  Logic-based model of breast cancer

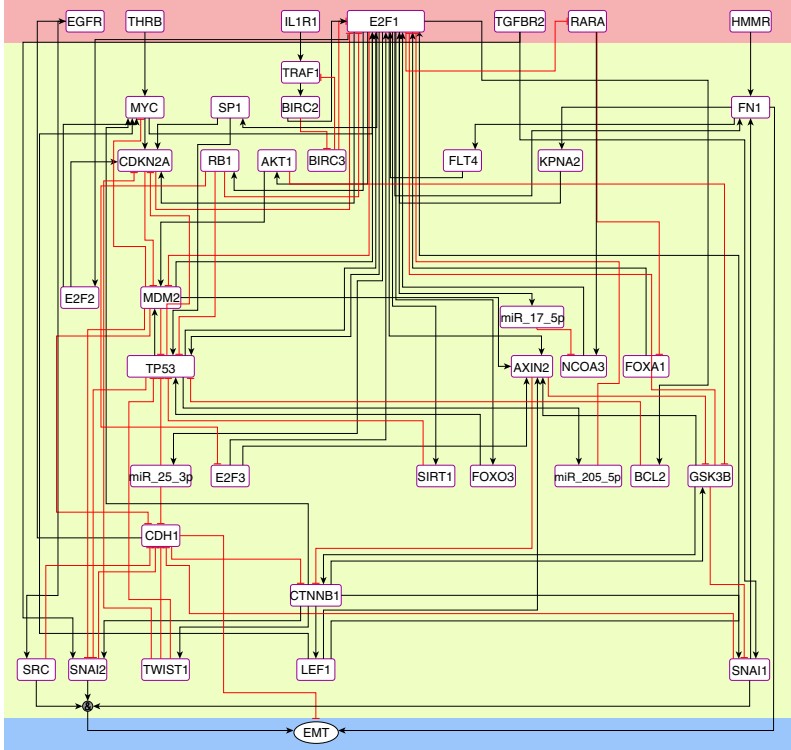

**Fig. 4** Logic-based models of the core network regulating invasive phenotypes in **a** bladder and **b** breast cancer. The *black* and *red lines* represent the type of interactions (i.e., activation and inhibition) among the interacting components. The microRNAs, receptors, and proteins are represented by *rectangular nodes*, whereas the EMT phenotype is represented by an *ellipse* in the output layer. The model is divided into three layers, i.e., the input layer (*pink*), the regulatory layer (*yellow*), and the output layer (*blue*)

(UM-UC-3, SW1710, J82, T24) and breast (MDA-MB231, BT549) vs. low expression in non-invasive or less-invasive epithelial bladder (RT-4, VM-CUB1, HT1197) and breast (MCF-7 and T47D) cell lines (Fig. 2a). Our experimental data are supported by gene expression data from the CCLE database (Supplementary Fig. 5). Increased expression of E2F1 commonly observed in EMT-like cell lines is also evident from several other aggressive tumor entities such as pancreatic, lung, and prostate cancer (Supplementary Fig. 6) as well as cutaneous melanoma[9]. Furthermore, overexpression of E2F1 induces an invasive phenotype in RT-4 and MCF-7 cells by upregulation of mesenchymal and downregulation of epithelial markers (Fig. 2b). In contrast, knockdown of the transcription factor in aggressive UM-UC-3 and MDA-MB231 cells results in reduced invasiveness and decreased expression of mesenchymal ZEB1 as well as upregulation of E-Cadherin in MDA-MB231 (Fig. 2c).

**Network motif prioritization**. From the E2F1 interaction map, we identified a large set of feedback loops ($n = 444$; 213 positive, 228 negative, and 3 neutral) that are responsible for non-intuitive behavior of the system (Supplementary Data 1). From such a large set of feedback loops, our aim was to identify the most important ones involved in the investigated phenotype. Towards this end, we used a weighted multi-objective function containing topological and non-topological network parameters (see "Methods"). Further, we selected different weighting scenarios for motif prioritization to avoid any bias induced by the parameters used in the multi-objective function (for details see Supplementary Methods).

We used KEGG's cancer disease pathway (KEGG: hsa05200) to estimate the number of nodes of a motif associated to a cancer pathway. For calculating the gene prioritization parameter, we first selected all the known EMT markers in our map proposed in Lanouille et al.[24] and calculated the score for all the nodes using a random walk with restart algorithm implemented in the Cytoscape plugin GPEC[25]. A complete list of the selected motifs along with structural and biomedical parameters as well as the motifs ranking scores are provided in Supplementary Data 1. Further, we selected the top ten motifs from each of the weighting scenarios implemented in the multi-objective function. In this way, we obtained 32 non-redundant motifs associated with an invasive phenotype in bladder and 28 with breast cancer, respectively.

**Derivation of tumor type-specific core regulatory network**. To obtain the core regulatory network, we merged all the unique motifs using the Cytoscape plugin NetDS[26] (v3.0). We retrieved three disjoint sub-networks in both tumor entities, which we connected by reviving interactions among the nodes from the E2F1 regulatory network. Thereby, we obtained tumor-specific (bladder and breast cancer) core regulatory networks (Fig. 3). Both networks include the transcription factors E2F1-3 and the cell cycle regulators RB1, MYC, CDKN2A, TP53/MDM2, SP1, FOXA1, FOXO3, and AKT1. Furthermore, both contain the enzyme SIRT1 that modifies targets like E2F1, TP53, and histones to silence their function. In addition, both core networks contain the CDH1 regulators SNAI1/2 and TWIST1, and interaction partners of CTNNB1 (AXIN2, LEF1).

In bladder cancer, we additionally found fibroblast growth factor receptor 1 (FGFR1) and its downstream regulatory subunits of the protein phosphatase 2 (PP2; inhibitor of cell growth and division), and the pro-proliferative inhibitor of PP2, KIAA1524 (CIP2A). Also, we found the transforming growth factor beta receptor (TGFBR) 1/2 downstream signaling molecules SMAD2-4 and their regulator ZEB1. In the breast cancer core network, we found Fibronectin 1 (FN1), which is related to

migration; FLT4 involved in angiogenesis; GSK3B, an anti-proliferative enzyme; KPNA2, a nucleopore transporter and the EP300-associated transcriptional activator NCOA3.

In addition, we observed feedback loops concerning nuclear factor kappa B subunit 1 (NFKB1) activation (CHUK, NFKBIA; related to cell survival) in bladder cancer, anti-apoptotic BIRC2/3, and pro-apoptotic TRAF1 factors in breast cancer, respectively. Interestingly, both loops are related to each other, as NFKB1 and BIRC2/3 are survival molecules and NFKB1 activates BIRC transcription. The regulatory cores, which we consider as the drivers of the invasive phenotypes contain 41 nodes and 107 interactions in bladder cancer and 35 nodes and 86 interactions in breast cancer.

**Logic-based models for EMT-driving molecular signatures**. To evaluate the input–output relationship of the obtained regulatory core networks, we used logic-based modeling formalism. The input layer of the logic-based models contains E2F1 and all receptors present in the regulatory core (Fig. 4). In bladder and breast cancer, two common receptors are part of the input layer: (i) EGFR and (ii) the retinoic acid receptor alpha (RARA). In addition, we found FGFR1 only in the regulatory core of bladder cancer. In simulations of the model, we considered signal propagation from input to output layer.

To capture all possible input signals to the regulatory network cores, we expanded the input layers by including additional receptors present in the comprehensive interaction network, which are directly connected to nodes constituting the regulatory cores. Thus, we included TGFBR, which is connected to SMADs, and the chemokine (C-X-C motif) receptor 1 (CXCR1) connected to ZEB1 and SNAI1 in the bladder cancer model. Similarly, we expanded the breast cancer input layer with the hyaluronan-mediated motility receptor (HMMR) connected to FN1; the TGFBR connected to SNAI1 and SNAI2; the interLeukin 1 receptor type I (IL1R1) and the thyroid hormone receptor beta (THRB) connected to TRAF1 and MYC, respectively. For bladder cancer, we selected TGFBR1 and for breast TGFBR2 due to their tissue-specific expression profiles. We derived Boolean functions for the input signals and their propagation through the nodes constituting the regulatory layer (Supplementary Data 2).

The output layer of the models comprises a unique node that represents the EMT process as the driver of the invasive phenotype, which is determined by a logic function involving the EMT markers present in the regulatory core. The output, determined through a multi-valued logic function, accepts four ordinal levels, ranging from 0 (no EMT) to 3 (high EMT).

**Predictive model simulations**. We determined the steady states of each variable in the model for different initial values of the input nodes. We consider two sets of scenarios, characterized by: (i) high expression of E2F1 (i.e., E2F1 = 1); and (ii) low expression (i.e., E2F1 = 0) in all possible Boolean combinations of receptors in the input layer. We obtained 64 input vectors for bladder cancer and 128 for breast cancer. Next, we simulated the network to determine the impact of the input vectors on the level of EMT (Table 1). Our simulation results suggest that when E2F1, TGFBR1, and FGFR1 are simultaneously active, bladder cancer cells become highly invasive (EMT = 3). A similar effect was observed in breast cancer when E2F1, TGFBR2, and EGFR are simultaneously active. Furthermore, we carried out in silico perturbation experiments to identify important nodes that can be exploited for therapeutic interventions. Perturbation experiments were performed for a highly invasive phenotype (EMT = 3) by changing the Boolean state of each node in the regulatory layer to reduce invasiveness. We used single and double perturbation

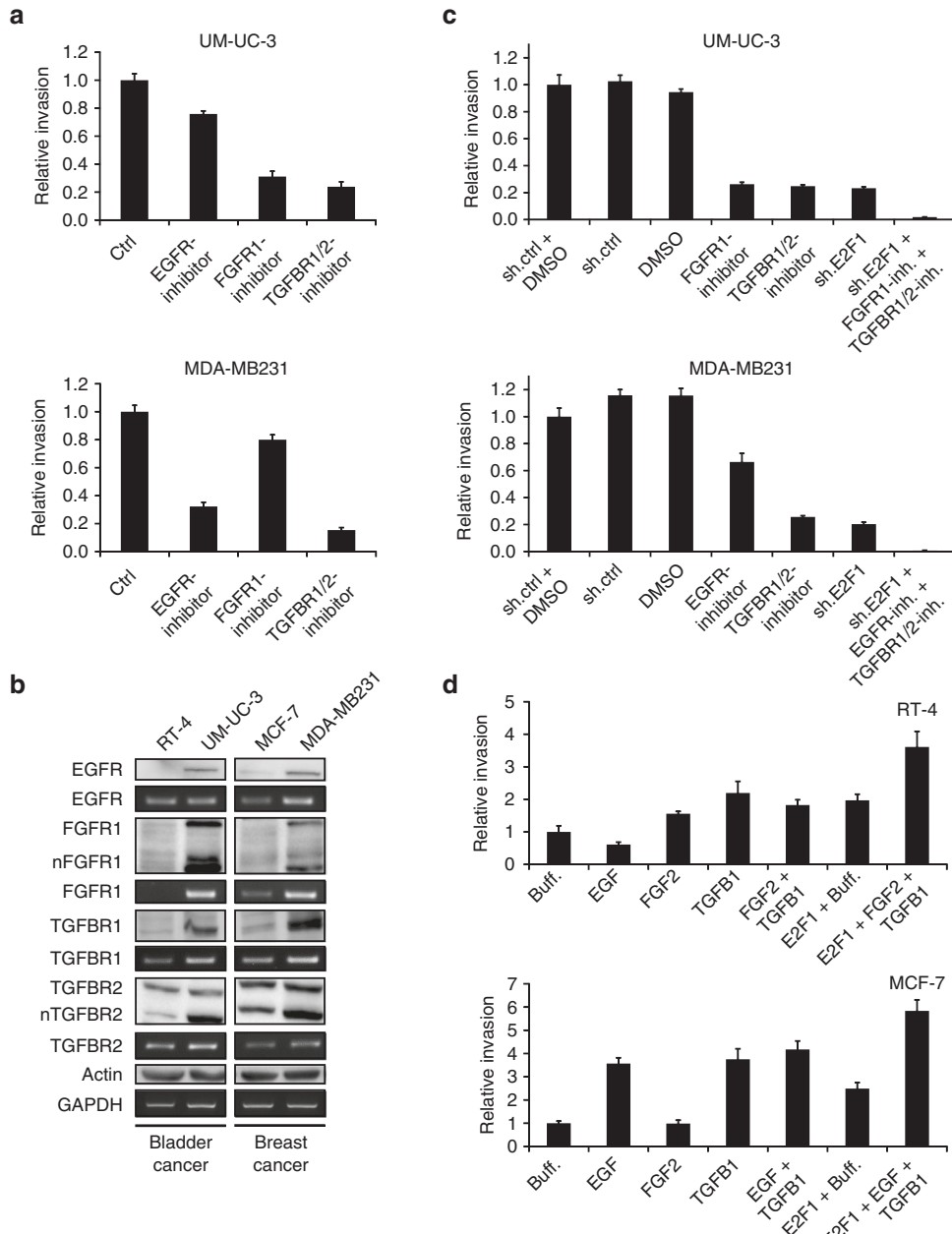

**Fig. 5** Effects of the EGFR, FGFR1, and TGFBR pathways on bladder and breast cancer invasion. **a** Different effects of EGFR inhibitor and FGFR1 inhibitor on invasive bladder (UM-UC-3) and breast cancer (MDA-MB231) cell lines. **b** Western blots and PCRs show the expression levels of the indicated receptors in less-invasive (RT-4, MCF-7) vs. invasive (UM-UC-3, MDA-MB231) cells. EGFR, FGFR1, as well as TGFBR1 are highly expressed in both invasive cell lines compared to the less-invasive ones. TGFBR2 expression levels equal within cell lines of each tissue type, whereas the nuclear fraction of TGFBR2 (nTGFBR2 as well as nFGFR1 for FGFR1) shows a clear upregulation in both invasive cancer cell lines. **c** Invasive potential of UM-UC-3 or MDA-MB231 was measured by Boyden chamber assay upon treatment with an *E2F1*-specific shRNA and inhibitors for TGFBR1/2, EGFR, or FGFR1 as indicated. **d** Boyden assay indicating the invasive potential of epithelial cell lines after stimulation with growth factors and/or overexpression of E2F1 transcription factor. Cells were preincubated in growth factor supplemented medium (10 ng/ml of EGF, FGF2, TGFB1, or both) and transduced with adenoviral vector for overexpression of ER-E2F1, which was activated by adding 4-OHT. Fold-changes were calculated relative to control cells (set as 1). All figures are representatives of at least three independent experiments. All *error bars* indicate s.e.m.

iteratively and observed the most prominent reduction of EMT in the latter case (simulation results are provided in Supplementary Data 5). Our perturbation results suggest that in bladder cancer (i) double knockout of *ZEB1* in combination with either *SNAI1*, *TWIST1*, *NFKB1*; (ii) knockout of *ZEB1* and activation of *CDH1*; or (iii) knockout of *SMAD2/3/4* in combination with *TWIST1* or *NFKB1* reduces EMT to 1. In case of breast cancer double perturbation by silencing *SRC*, *FN1*, *SNAI1*, *SNAI2*, or

activation of *CDH1* in any of the combinations reduces EMT to 1 (Supplementary Table 4).

**Validation of in silico predictions with cell line models**. Our model predictions revealed a common impact of E2F1 and TGFB1 signaling on tumor invasiveness in both cancer types. More specifically, TGFBR1 and FGFR1 in combination with

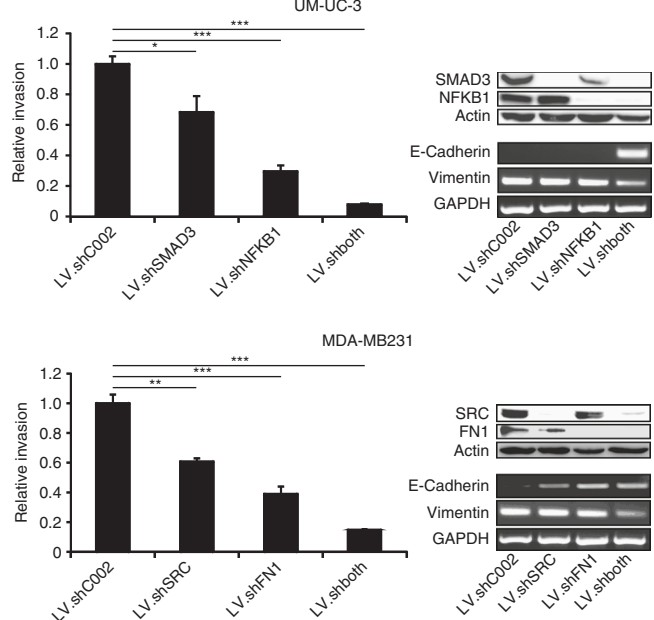

**Fig. 6** Validation of in silico knockout simulations by gene knockdown experiments. By in silico simulations, we identified the most effective combination of double knockouts regarding reversal of EMT. The invasive potential of UM-UC-3 and MDA-MB231 was measured by Boyden chamber assay upon treatment with lentiviral vector expressing *SMAD3*, *NFKB1*, *SRC*, or *FN1*-specific shRNA, alone or in combination (*left panels*). Changes of EMT markers *E-Cadherin* and *Vimentin* after gene knockdown are shown on transcriptional level (*right panels*). Fold-changes were calculated relative to control cells (treated with control LV.shC002; set as 1). All *error bars* indicate s.e.m. For statistical significance *t*-test was used (\*P-value < 0.05; \*\*P-value < 0.01; \*\*\*P-value < 0.001). ShRNA efficiency was confirmed by western blot

highly expressed E2F1 induce the most invasive phenotype in bladder cancer, whereas in breast cancer, it is the combined action of TGFBR2, EGFR, and E2F1 that triggers high levels of invasiveness. To validate the predicted influence of the receptors and E2F1 on the invasive phenotype, we used chemical inhibitors and a shRNA-based approach to target these key players. In line with the in silico simulations, inhibition of EGFR in UM-UC-3 (bladder) or FGFR1 in MDA-MB231 (breast) had a minor influence on cell invasion (Fig. 5a). To confirm that this is not due to a lack of EGFR in UM-UC-3 or FGFR1 in MDA-MB231, respectively, receptor expression was confirmed by PCR and immunoblot in all cell lines. As Fig. 5b shows, all receptors are highly expressed in both invasive cell lines. Furthermore, as predicted by the simulations, inhibition of E2F1, TGFBR1/2 and FGFR1 in UM-UC-3 and E2F1, TGFBR1/2, and EGFR in MDA-MB231 had a tremendous impact on the invasive behavior of the respective cell line with the highest effect upon combined inhibition (Fig. 5c).

Referring to the initial data where we have shown induction of an invasive phenotype in epithelial cell lines (RT-4, MCF-7) by overexpressing E2F1, we now stimulate the signaling pathways by applying their respective ligands (epidermal growth factor (EGF), fibroblast growth factor 2 (FGF2), transforming growth factor beta-1 (TGFB1)) to induce invasion in those cell lines. Figure 5d demonstrates that the in silico predicted selective response to the different stimuli actually occurs: In RT-4 cells, overexpression of E2F1 or stimulation with FGFR1 ligand or TGFBR2 ligand (alone or in combination) raises the invasive potential, whereas stimulation with EGFR ligand has no effect on cell invasion. In

contrast, EGFR stimulation of MCF-7 cells promotes cell invasion in a manner comparable with the TGFBR1/2 stimulation or E2F1 overexpression. Here, activation of FGFR1 has no impact on the invasive potential of the MCF-7 cell line. Taken together, by combining the predictions from in silico simulations and the in vitro experimental validation, we were able to find molecular signatures that regulate invasive phenotypes in E2F1-driven bladder and breast cancer.

To validate findings of the in silico perturbation simulations, we decided to knockdown *NFKB1* and *SMAD3* in UM-UC-3, and *SRC* and *FN1* in MDA-MB231, instead of modulating the other well known EMT markers *SNAI1/2*, *TWIST1*, *ZEB1*, or *CDH1*. We performed single and double knockdown of these genes and measured both the transcriptional and EMT/MET response. Although the removal of single genes resulted in a clear reversal of the EMT phenotype (reduced invasion) in both cell lines, the strongest effect was observed after double knockdown, as demonstrated by their lowest invasive capacity, increased E-Cadherin, and decreased Vimentin levels (Fig. 6).

**Validation of model predictions with patient data**. To further validate the molecular signature that regulates invasiveness in bladder cancer, we used data from a patient cohort (*n* = 165)[14], in which a correlation between *E2F1* expression and superficial to invasive progression was observed (GEO id: GSE13507). We grouped the patients into high and low expression profiles of *E2F1*, *TGFBR1*, and *FGFR1* from their respective median expression values. For each group, we calculated the progression-free survival probability and found that the survival probability was higher in the patient group with low expression of each molecule individually. Furthermore, we identified the subgroups of patients with high vs. low expression of: (i) *E2F1–FGFR1*; (ii) *E2F1–TGFBR1*; and (iii) *E2F1–FGFR1–TGFBR1*. The progression-free survival probability of each subgroup reveals that patients with high expression of *E2F1–FGFR1* have lowest mean survival time (33.79 months), whereas those with low expression of *E2F1–FGFR1–TGFBR1* have the best prognosis (93.35 months) among all the subgroups analyzed (see Kaplan–Meier plots in Fig. 7a–c). Our analyses indicate that patients with low expression survive more than twice as long as the patient subgroup with high expression of the molecular signature.

To validate the molecular signature from the regulatory core in breast cancer, we used data from the TRANSBIG network (GEO id: GSE7390; *n* = 198) generated by Desmedt et al.[22] Similar to bladder cancer patients, we observed that the progression-free survival probability of breast cancer patients was low for high expression of *E2F1*, *EGFR*, and *TGFBR2* in different combinations (Fig. 7d–f). Interestingly, we observed the highest mean survival time (97.29 months) in the patient subgroup with low expression of *E2F1–EGFR–TGFBR2*. Similar to the bladder cancer analyses, the patient subgroup with high expression of all three components had nearly half the mean survival time (50.11 months) compared with the subgroup with low expression.

We further validated molecular signatures in large patient cohorts of TCGA bladder cancer (BLCA; *n* = 426) and TCGA breast cancer (BRCA; *n* = 1218) accessible through UCSC Xena (http://xena.ucsc.edu). We found that signatures predicted using Boolean simulations were able to distribute patients into early vs. advanced stages in bladder cancer and aggressive vs. less-aggressive stages in breast cancer significantly (*P*-value < 0.005) (Fig. 8a, b; Supplementary Fig. 7). To assess the capability of our workflow in predicting significant molecular signatures associated with invasive phenotypes, we generated 30 random signatures of three nodes from each of the regulatory cores and arbitrarily

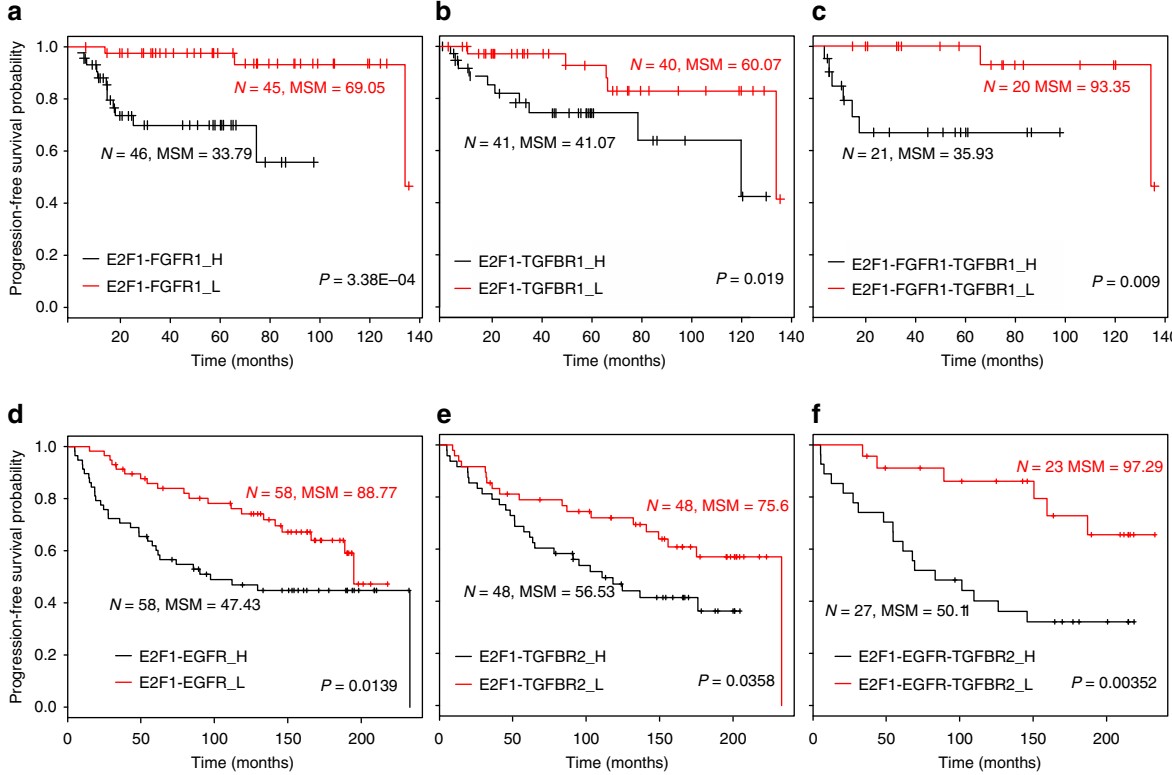

**Fig. 7** Kaplan–Meier plots of progression-free survival of patients with bladder and breast cancer. Plots **a–c** show the survival curves for patients with high and low expression of combined signatures (*E2F1–FGFR1*; *E2F1–TGFBR1*; and *E2F1–FGFR1–TGFBR1*) in bladder cancer patients, whereas plots **d–f** are for the combined signatures of *E2F1–EGFR*; *E2F1–TGFBR2*; and *E2F1–EGFR–TGFBR2* in breast cancer patients. In both cases, patients with low expression of the molecular signatures have high mean survival times and vice versa. High expression of molecular signature(s) is represented as '_H' (*black curve*) and low expression as '_L' (*red curve*). '*N*' is the number of patients observed with high/low expression of molecular signatures and 'MSM' is the mean survival month from the patient group. *P*-values shown in the figures are for log rank test

assigned high or low expression values (Supplementary Data 7 and 8). We observed that the molecular signature predicted for bladder cancer is the only one that nicely distinguishes between the early and advanced stage of disease (Fig. 8c). In case of breast cancer, in addition to the predicted signature, some of the random signatures were also able to distinguish between aggressive and less-aggressive cancer types (Fig. 8d). This might be due to the highly heterogeneous nature of breast cancers. Overall, our analysis reveals that an invasive tumor phenotype in bladder cancer is driven by *E2F1*, *TGFBR*, and *FGFR1*, whereas in case of breast cancer it is driven by *E2F1*, *TGFBR*, and *EGFR*.

## Discussion

To improve the treatment outcomes of patients who develop metastases and drug resistance, a mechanistic understanding of the determinants of these processes is indispensable. A large number of clinical studies have recently been published which identified E2F1 as a key transcription factor that switches duties from a tumor suppressor to a driver of metastasis[11, 14, 27]. To understand how E2F1 switches its duties, we derived a comprehensive interaction map (Fig. 1) that includes state of the art knowledge on transcriptional, post-transcriptional, and protein–protein interactions around the E2F family. The map contains 879 nodes and 2278 interactions of gene regulation and signaling processes associated with the E2Fs, thereby providing ways not only for the detailed elucidation of E2F regulation but also for tracing their connections to other cancer-related pathways. Recently, the idea of analyzing regulatory maps by integrating multi-omics data for the detection of disease driving

molecules and the identification of therapeutic targets has gained momentum[28–32]. In the context of our work, Calzone et al.[28] reconstructed a comprehensive map of the E2F transcription factor family. Their work focused on the differing roles of E2F family members in the cell cycle, reflecting the complex interplay between the E2Fs, RB1, its homologs RBL1 and 2, the cyclins/cyclin-dependent kinases, and cell cycle arresters. In contrast, our map sets a main focus on the newly discovered role of activating members of the E2F family (E2F1-3) in cancer development and progression, with an emphasis on pro-apoptotic and anti-apoptotic (survival), angiogenic as well as functions relevant for EMT. We included additional key players connected to E2F1 directly or through its neighbors along with a post-transcriptional layer of microRNAs in the context of cancer. Interestingly, the majority of the components in the map by Calzone and coworkers are included in our map (see Supplementary Fig. 8 for further details).

The underlying idea of our approach is that the structural and data-driven analysis of this map allows the identification of key functional modules, here named core regulatory networks, composed of regulatory motifs and critical molecular interactions that drive given cancer phenotypes. To this end, we have integrated coherent workflow tools coming from data analysis, bioinformatics, and mathematical modeling. Precisely, and to the best of our knowledge, we do not find in the literature a precedent of combining network-based high-throughput data analysis, network reduction, and Boolean modeling. Existing work either focuses on network-based analysis[5, 33] or Boolean network construction and simulation[34, 35]. Further, our methodology includes an innovative element in terms of network reduction,

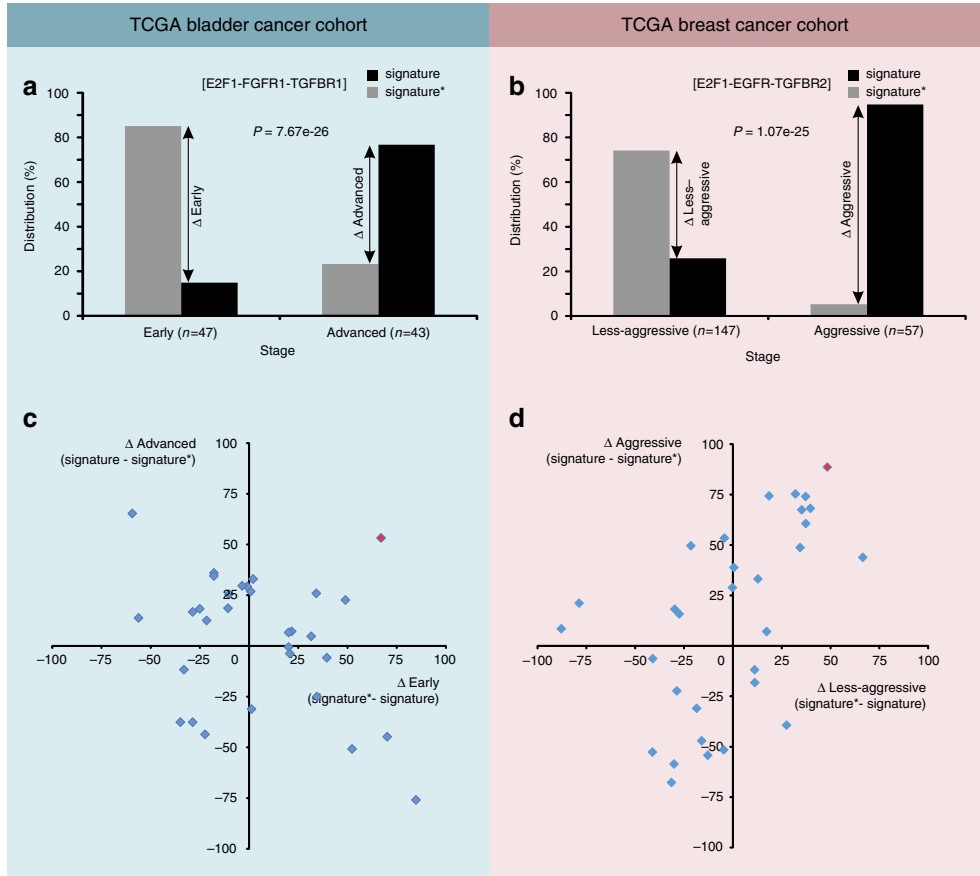

**Fig. 8** Validation of molecular signatures in **a** TCGA bladder and **b** TCGA breast cancer cohorts. In the bladder cancer cohort, we first filtered patients diagnosed with early (stage II) and advanced (stage IV) stages. Further in both stages, we identified subsets of patients where expression of genes of the molecular signature (i.e., E2F1, TGFBR1, and FGFR1) was above (signature) or below (signature*) the respective mean expression value. In case of the breast cancer cohort, we classified patients into aggressive (basal & Her2) vs. less-aggressive (luminal A and B) molecular subtypes based on PAM50 stages provided. Afterwards we identified those subsets of patients where expression of genes of the molecular signature (i.e., E2F1, TGFBR2, and EGFR) was above (signature) or below (signature*) the respective mean expression value. We calculated distribution difference (%) of patients within pathological stages in both cancer types. Similarly, distribution differences for 30 random signatures vs. signatures* derived from **c** bladder and **d** breast cancer regulatory cores are shown as scatter plots. Signatures in the upper right or lower left corners nicely distinguish patient phenotypes. The molecular signatures predicted from our workflow are shown in red. P-value is calculated using Pearson's $\chi^2$ test

namely the use of an algorithm employing multi-objective optimization concepts to rank and select key regulatory motifs, based on network topology features and expression profiles. As far as we know, this has not been explored before in the context of cancer.

The analysis of topological properties can provide important information about the cues that have a significant impact on the dynamics of the network[7, 36]. In our workflow, we analyzed the network properties node degree and betweenness centrality. Nodes with a high degree, often named hubs, are known for their importance in network organization and very often are transcription factors having a central role in orchestrating cell differentiation programs[37, 38]. Nodes with high betweenness centrality serve as gate keepers in the communication between different components of a network[39]. As our network was constructed with all possible regulatory processes around E2F1, one can expect that values of the topological properties for some of the nodes are relatively higher than for others, a potential bias compensated by also considering non-topological properties. Precisely, we assigned different weights to the genes in the network according to their known relatedness to relevant cancer-associated pathways. Furthermore, we used the gene expression fold-change in cell lines reflecting the cancer phenotypes

investigated, thereby providing a data-driven approach to make the core network cancer-type and context-specific.

As sets of genes involved in certain phenotypes are highly interconnected and regulate each other through coherent and incoherent regulatory loops (motifs) from different pathways, analysis of these can provide key insights into the structure and dynamics of the network[40] followed by identification of disease biomarkers[41, 42]. In intracellular regulatory networks, feedback loops provide stability and robustness against intrinsic and extrinsic noise, homeostasis or even all-or-nothing patterns of activation[6, 40, 43]. Very often, these network motifs are disrupted or abnormally regulated in cancer and therefore, the analysis of their differential regulation provides important information on the emergence of cancer phenotypes. However, the identification of important feedback loops in a highly connected network is a methodological challenge. From the E2F1 interaction map, we identified a large set of three-nodes feedback loops responsible for non-intuitive behavior of the system (Supplementary Data 1). We considered three-node feedback loops due to the fact that larger sized network loops are typically composed of one or more three-node loops[43].

Motif identification-based methods have been previously used to recognize key network regulators. For example, Zhang et al.[41]

ranked network motifs using gene expression data to detect breast cancer susceptible genes and Koschützki et al.[44] used motifs with various network topological parameters to identify important nodes in a biochemical network. We here introduced a new motif ranking scheme using a weighted multi-objective function that integrates topological (e.g., node degree and betweenness centrality) and non-topological (e.g., gene expression, gene prioritization) properties. Topological properties account for the structural importance of the nodes, and non-topological properties for their cancer-type and context-specific relevance. We used multiple weighting scenarios in the multi-objective function to provide motif ranking as unbiased as possible regarding the properties assessed in the function. Our proposed multi-objective function and ranking scheme can easily be extended to add new information layers in the workflow. Thus, we think that the method proposed can be used for investigating other cancer networks besides those focused on the E2F family discussed here.

Using our proposed multi-objective function for motif prioritization, we have ranked feedback loops identified in the E2F1 interaction map (Supplementary Data 1). To understand the combined effect of top-ranked feedback loops on the regulation of EMT processes in bladder and breast cancer, we interconnected them to generate tumor-specific regulatory core networks (Fig. 3), which we believe are the main drivers of the network dynamics. Furthermore, we analyzed the core regulatory network for the identification of molecular signatures driving the invasive phenotype by using logic-based model simulations[45]. We used Boolean logic for the input and regulatory layers, whereas multi-valued logic representation for the phenotypical output of the network. Multi-valued logic allows us to model several activity levels of the phenotype, which helps assessing the aggregated effect of various network components on the phenotype[29]. However, the use of multi-valued logic increases the complexity of the model, and therefore, we apply it only to the phenotypical output.

The in silico simulation results indicated that high levels of E2F1–TGFBR1–FGFR1 in bladder cancer and E2F1–TGFBR2–EGFR in breast cancer constitute a molecular signature that represent the most aggressive phenotype. Surprisingly, the other receptors that are part of the input layers in the models had no effect on the EMT process in our simulations. Our results are in agreement with previous experimental findings in bladder cancer studies where high levels of E2F1[14], TGFBR1[46], and FGFR1[47] were independently associated with tumor invasion. Similarly, in breast cancer studies, high expression of E2F1[48], TGFBR2[49], and EGFR[50] was separately observed to regulate invasive tumor phenotypes. Furthermore, we confirmed the role of predicted signatures on the regulation of tumor invasion using bladder and breast cancer patient survival data from independent studies in Figs 7 and 8. For all our predicted signatures high expression of the constituent molecules mapped to low patient survival and vice versa. These correlations prove that our approach is successful in identifying tumor-specific molecular signatures regulating EMT processes and driving invasive phenotypes.

By applying an expression signature from highly invasive bladder cancer cells and the respective regulatory core, our model predicted resistance to EGFR inhibition in cells overexpressing E2F1. Indeed, treatment of UM-UC-3 with EGFR inhibitors resulted only in a marginal reduction of cell invasion. Likewise, exposure of RT-4 to EGF did not have any observable effect. These results are not intuitive as we showed that UM-UC-3 express EGFR (Fig. 5b). On the basis of these results, we propose that EGFR-targeted therapies might be ineffective in muscle invasive bladder cancer exhibiting elevated levels of E2F1. In this

regard, it is interesting to note that the majority of completed clinical trials using inhibitors of EGFR family RTKs do not show an added benefit over standard of care chemotherapy in an adjuvant or second line setting[51]. These studies show that treatment with chemotherapeutic agents rendered patients with muscle invasive bladder cancer also resistant to EGFR inhibitors for hitherto unknown reasons. Expression of EGFR is, like in our model system UM-UC-3, detected in urothelial carcinomas of the bladder. However, the absence of activating EGFR mutations at exons 19 to 21 in a number of bladder cancer specimens has been demonstrated and may contribute to EGFR inhibitor resistance[52]. In contrast, aberration of FGFR1 is a frequent event in bladder cancer contributing to rapid disease progression[53]. In pre-clinical models, the presence of FGFR1 genetic alterations confers sensitivity to FGFR inhibitors[54]. Respective clinical trials are ongoing. However, molecular pre-selection (e.g., FGFR1 amplifications present or not) is the primary challenge for the development of FGFR inhibitors. In this context, our model could be a valuable tool to select for patients who potentially benefit from an anti-FGFR therapy through application of model-based signatures for patient classification such as determination of E2F1 and TGFBR1 oncogene expression. With regard to our data in bladder cancer cell lines, we observed an opposite situation in breast cancer cells. Although EGFR activation in less-invasive MCF-7 or inhibition in metastatic MDA-MB231 showed a clear regulation of cell invasion, FGF treatment, and exposure to FGFR1 inhibitors, respectively, did not substantially alter cell invasion. However, the FGFR signaling pathway regulates normal mammary gland development and FGFR1 overexpression has been associated with breast cancer progression[55]. A recent study applied a pre-clinical mammary tumor model to show that FGFR1 inhibitor treatment leads to initial rapid regression which is, nevertheless, finally followed by tumor recurrence[56]. Intriguingly, recurrent tumor tissues revealed elevated levels of activated EGFR compensating for FGFR1 inhibition. It remains to be seen whether or to which extent available FGFR1 inhibitors can improve treatment of metastatic breast cancer[57].

Overall, model-based treatment recommendations of E2F1-driven tumor diseases such as advanced bladder or breast cancer, have the potential to support cohort-specific treatment of patients to avoid therapy resistance and cope with aggressive cancers. Finally, we used the map to investigate E2F1-associated malignant progression in two tumor entities, but the map and the workflow proposed can also be applied to other cancer types in which E2F1 might have a similar role, as well as to uncover other phenotypes related to this transcription factor like chemoresistance or angiogenesis.

## Methods

**Data retrieval and construction of the E2F1 interaction map.** For the construction of the E2F1 interaction map, we derived, curated, and incorporated information from the literature and publically available databases as well as E2F1 cofactors recently identified by our group. More specifically, we retrieved data on protein–protein interactions from STRING[58] (v9.1) and HPRD[59] (release 9). Transcription factors and their target genes were retrieved from databases[60] and relevant literature. Moreover, we included microRNA-target interactions, which were extracted from the miRTarBase[61] database (release 4.5). Transcription factors of microRNAs have been extracted from the TransmiR[62] database (v1.2). In addition, we searched PubMed for publications about validated E2F transcription factors, their post-translational modifications, molecular interactions, and connections to certain diseases, especially to cancer. Furthermore, we manually curated interactions (assigned directions to the interactions, i.e., activation/inhibition and relevant references) that were retrieved from mostly automatically generated databases like STRING, KEGG or the EMBL-EBI search engine PSICQUIC[63] and the text mining tool iHop[64] to search for more interactions described in the literature.

The E2F1 interaction map was built with the process diagram editor CellDesigner[65] (v4.3) and visualized as a SBGN (Systems Biology Graphical Notation) compliant diagram[66]. We created different regulatory and functional

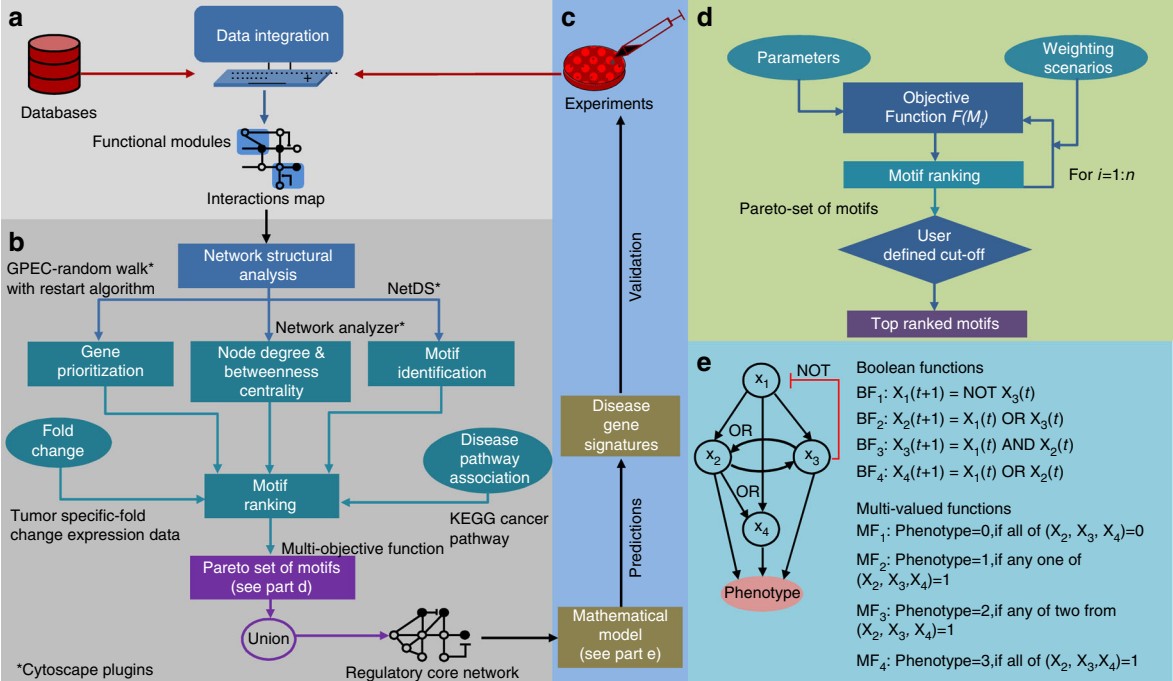

**Fig. 9** Methodology for the construction of an interaction map, derivation of the regulatory core, and prediction of molecular signatures using model simulations. **a** Data on molecular interactions are extracted from public databases and the literature to reconstruct the regulatory network underlying the cancer phenotype investigated. **b** Topological and non-topological network properties are used to identify important network motifs. A regulatory core is derived by merging the top-ranked motifs. **c** In silico simulations of the regulatory core help in the detection of molecular signatures that are then subject to experimental validation. **d** An algorithm for motif prioritization. Motifs can be prioritized based on the objective function represented in Eq. (1). This objective function contains parameters accounting for node properties, the expression profile of nodes in relevant cancer cell lines and their relatedness to disease pathways. Weights can be assigned by giving importance to a particular parameter in a user defined manner or iteratively to determine the Pareto sets of motifs. From these sets, top-ranked motifs can be identified based on user defined cutoff. **e** Logic-based representation of biochemical network. *Left*: A toy model of biological network consisting of four nodes ($X_{1-4}$), and interactions among them regulate certain phenotype. *Right*: Derivation of Boolean functions (*BF*) and multi-valued logic functions (*MF*) for the toy model

compartments based on the role of molecules in influencing E2F1 activities and determining cell fate as follows: (i) Extra-/intracellular receptor signaling: This compartment contains cellular receptors and ligands with known crosstalk to E2F family pathways (e.g., FLT4, PDGFRB) and additionally, cancer-relevant receptors that feed downstream processes like the cell cycle (e.g., growth factor receptors via the MAPK pathway), apoptosis (like TNF/TRAIL receptors) or survival (e.g., IGF1R for Ras and AKT signaling); (ii) E2F1 modifications: To separate the post-translational modifications influencing its transactivation potential of target genes from the effects that cofactors have on the latter, we consider post-translational modifiers of E2F1, containing factors that regulate E2F1 protein stability, for example, upon phosphorylation after DNA damage, including kinases (ATM/R), acetylases (EP300), deacetylases (HDAC1) or methyltransferases (DNMT1), and regulators of E2F1 transcriptional activity, containing protein-binding partners that regulate the affinity or specificity of E2F1 to its DNA targets (like epigenetic modifiers interacting with histones and recruiting E2F1 to target promoters, e.g., ATAD2); (iii) Cell cycle: This compartment includes cyclins, CDKs, and MYC that regulate the cell cycle upon extracellular stimulation by growth factors. It also harbors the regulation of expression and activity (e.g., inhibition by RB1) of E2F1-3 as central cell cycle regulators; (iv) Quiescence: This compartment encloses factors involved in arresting the cell cycle (CDK inhibitors such as p21, p14/ARF), as well as complexes that silence E2F1 targets during G0/G1/G2-phase or quiescence (e.g., SWI/SNF complex or the DREAM complex).

DNA repair: DNA damage sensing and repair factors (e.g., BRCA1, TOPBP1) are summarized in this compartment; (v) Apoptosis: Factors inducing and executing the cellular apoptotic program like the TP53 family (TP53/63/73), pro-apoptotic BCL-family inhibitors (BID, BAX), or Caspases; (vi) Survival: Factors suppressing pro-apoptotic signaling like the anti-apoptotic BCL-family members BCL2, MDM2, XIAP proteins, MDR-transporters, and the MYB family; (vii) EMT/invasion/angiogenesis: Besides currently known players in EMT-like ZEB1/2, SNAI1/2, VIM, CDH1/2, this compartment contains the CTNNB1 regulation and its influence on CDH1 as well as angiogenesis and extracellular matrix regulating factors such as HIF1A, MMPs, and L1CAM; (viii) Metabolism: Ion channels (KCNH11) and metabolic enzymes (PFKFB2, MAT2A) are summarized in this compartment as there is recent evidence indicating a regulatory role for E2Fs in the cellular metabolome.

Next, we incorporated into our map detailed text-based annotations including HUGO name, HGNC ID, Entrez ID, and UniProt ID. We used complex formation

and dissociation information to accommodate activities of protein monomers, dimers, and oligomers (Supplementary Data 3). By assigning web links to the annotations in our map, we turned it into an interactive resource.

The web version of the E2F1 map was constructed using NaviCell[67]. In addition, the map was translated to Cytoscape[68], a format suitable for the use of network analysis tools. For the sake of improving the visualization, the CellDesigner map (Fig. 1) pools proteins of the same family into general terms, whereas the Cytoscape version (Supplementary Fig. 1) contains all respective family members and interactions among them (Supplementary Data 4), thereby allowing the use of existing tools for data integration and network analysis. To assure the accuracy of the network, we randomly selected ~10% of the interactions and asked independent domain experts to cross-validate them. Over 98% of the interactions were derived correctly.

**Identification of context-specific regulatory core network**. We developed a novel method for the identification of the tumor entity-specific core of large regulatory networks, which we understand as a subnetwork that is responsible for critical systems dynamics and driver of a tumor-specific phenotype. Furthermore, we propose a mathematical modeling-based approach for the prediction of molecular signatures, i.e., sets of network-derived diagnostic/prognostic biomarkers. For an illustration of the workflow for network reconstruction, identification of the regulatory core, and prediction of molecular signatures see Fig. 9.

The method for the identification of the regulatory core and molecular signatures involves the following steps: (i) Network analysis: The purpose is to identify the network structure and node properties, which help in the identification of important nodes and network motifs. Topological and node properties were determined using the Cytoscape plugin NetworkAnalyzer[69]. In particular, we calculated for each node the degree and betweenness centrality, and for the network the clustering coefficient, diameter, radius, characteristic path length, and average number of neighbors to understand the overall organization of the network (Supplementary Table 1; Supplementary Fig. 2). (ii) Network motif identification: Feedback loops from the E2F1 interaction map were identified using Cytoscape plugin NetDS[26] (v3.0). For the identification of feedback loops, we set the loop length to three nodes (see "Discussion" section for detail). (iii) Motif ranking: To identify the most important motifs with respect to the relevance for the disease

### Table 1 The effect of E2F1 and receptor molecules in relevant combinations on the EMT phenotype in bladder and breast cancer model

**(a) Bladder cancer**

| E2F1 | TGFBR1 | FGFR1 | EGFR | CXCR1 | RARA | EMT |
|---|---|---|---|---|---|---|
| 0 | 0 | 0 | 0/1 | 0/1 | 0/1 | 0 |
| 0 | 0 | 1 | 0/1 | 0/1 | 0/1 | 1 |
| 0 | 1 | 0 | 0/1 | 0/1 | 0/1 | 1 |
| 0 | 1 | 1 | 0/1 | 0/1 | 0/1 | 2 |
| 1 | 0 | 0 | 0/1 | 0/1 | 0/1 | 1 |
| 1 | 0 | 1 | 0/1 | 0/1 | 0/1 | 2 |
| 1 | 1 | 0 | 0/1 | 0/1 | 0/1 | 2 |
| 1 | 1 | 1 | 0/1 | 0/1 | 0/1 | 3 |

**(b) Breast cancer**

| E2F1 | TGFBR2 | EGFR | HMMR | THRB | IL1R1 | RARA | EMT |
|---|---|---|---|---|---|---|---|
| 0 | 0 | 0 | 0/1 | 0/1 | 0/1 | 0/1 | 0 |
| 0 | 0 | 1 | 0/1 | 0/1 | 0/1 | 0/1 | 1 |
| 0 | 1 | 0 | 0/1 | 0/1 | 0/1 | 0/1 | 1 |
| 0 | 1 | 1 | 0/1 | 0/1 | 0/1 | 0/1 | 2 |
| 1 | 0 | 0 | 0/1 | 0/1 | 0/1 | 0/1 | 1 |
| 1 | 0 | 1 | 0/1 | 0/1 | 0/1 | 0/1 | 2 |
| 1 | 1 | 0 | 0/1 | 0/1 | 0/1 | 0/1 | 2 |
| 1 | 1 | 1 | 0/1 | 0/1 | 0/1 | 0/1 | 3 |

Active state of the molecule is represented by '1' and the inactive state as '0'. The phenotype output (EMT) can take four ordinal levels ranging from '0' (non-invasive) to '3' (highly invasive). Table (a) is the summary of 64 in silico simulations of bladder cancer. Each row represents the result of eight simulations where for the given Boolean state of E2F1, TGFBR1, and FGFR1, all eight combinations of EGFR, CXCR1, and RARA results in the same phenotypical output. Table (b) is the summary of 128 in silico simulations of breast cancer. Each row represents the result of 16 simulations where for the given Boolean state of E2F1, TGFBR2, and EGFR, all 16 combinations of HMMR, THRB, IL1R1, and RARA results in the same phenotypical output

phenotype under investigation, we developed a novel motif ranking scheme (Fig. 9d). The scheme is based on: (a) Topological properties, node degree and betweenness centrality (see "Discussion" section for detail); (b) the involvement of the motif constituents in KEGG's 'Pathways in cancer' pathway (KEGG: hsa05200); (c) the gene prioritization score from the Cytoscape plugin GPEC[25]; and (d) tumor-specific gene expression fold-changes from non-invasive to invasive phenotypes. We used the ArrayExpress database (ArrayExpress accession number: E-MTAB-2706)[70] to find suitable gene expression data in non-invasive and invasive bladder and breast cancer cell lines. In particular, we used RT-4 as non-invasive and UM-UC-3 as invasive cell lines in bladder cancer, whereas MCF-7 as non-invasive and MDA-MB231 as invasive cell lines in breast cancer. Differential expression analysis was performed using the DEseq R-package with *method = 'bind'* and *fitType = 'local'* (v1.22.1)[71]. Furthermore, we calculated the absolute average fold-change for a motif based on the change in expression values of each node in non-invasive to invasive phenotype. To rank the network motifs considering all these structural and biomedical criteria, we derived the weighted multi-objective function[72] in Eq. (1).

$$S_{ij} = \frac{w_{1j}}{2} \cdot \frac{\langle ND \rangle_i}{\max(ND)} + \frac{w_{1j}}{2} \cdot \frac{\langle BC \rangle_i}{\max(BC)} + w_{2j} \cdot \frac{\langle DP \rangle_i}{\max(DP)} \\ + w_{3j} \cdot \frac{\langle GP \rangle_i}{\max(GP)} + w_{4j} \cdot \frac{\langle |FC| \rangle_i}{\max(|FC|)}$$

(1)

Here, $S_{ij}$ is the ranking score of each motif ($i = 1…n$) in different weighting scenarios ($j = 1…13$) as given in Supplementary Table 2. $w_{1j}$ to $w_{4j}$ are weighting factors pounding the importance of the chosen properties, $\langle ND \rangle_i$: average node degree, $\langle BC \rangle_i$: average betweenness centrality, $\langle DP \rangle_i$: number of nodes in a motif involved in disease pathways, $\langle GP \rangle_i$: average gene prioritization score, and $\langle |FC| \rangle_i$: average absolute expression fold-change of a motif $i$.

To give equal importance to each property, the function is normalized to the maximum property value in all the network motifs identified (e.g., max(BC)). In order to not over-emphasize topological properties in motif prioritization, we assigned half of the weighting factor to $\langle ND \rangle$ and $\langle BC \rangle$. To generate a ranking of the motifs, we computed the value of the $S_{ij}$ function for every motif $i$ identified. The function proposed is intrinsically multi-objective and may generate a different ranking for same motif depending on the sets of values chosen for the weighting factors (Supplementary Methods). To approximate the Pareto set of all non-dominated motif rankings, we iteratively modified the values of the weighting factors, computed the $S_{ij}$ function for every motif and ranked them according to

their $S_{ij}$ values (Supplementary Data 1). Next, we selected the top 10 motifs from each weighting scenario for further analysis (Fig. 9d). (iv) Derivation of the regulatory core: The core regulatory network is obtained by merging the sets of the top-ranked motifs identified using Cytoscape plugin NetDS (v3.0). In case there are disjoint sub-networks in a regulatory core, we connect those using direct interactions between the nodes taken from the complete regulatory network. (v) Prediction of molecular signatures using in silico simulations: We consider a disease gene signature as a group of molecular entities in the regulatory core, which upon perturbation have a significant impact on the disease phenotypes. To identify molecular signatures, the regulatory core is translated into a logic-based model[29, 73] and steady state analysis is performed using the software tool CellNetAnalyzer[74].

We developed logic-based models of the regulatory cores and carried out in silico perturbation experiments. To this end and upon the selection of the relevant network motifs, we established the Boolean rules based on the network structure and the inspection of the available literature about the interactions. The obtained model contains three layers: (i) An input layer, (ii) A regulatory layer, and (iii) An output layer representing the phenotype. In the Boolean-like logic models, nodes $X = (X_{1…n})$ of a network correspond to the Boolean variables that can have values either 1 or 0, and edges define the type of interactions (e.g., activation or inhibition) that can be represented by Boolean gates (ACTIVE, NOT, OR, and AND). In Boolean models, the future state ($t + 1$) of a node is a Boolean function (BF) of the current state ($t$) of all the nodes regulating it, i.e., $X_i(t + 1) = BF(X_1(t), X_2(t),…, X_n(t))$ (Fig. 9e). We derived Boolean functions for signals originating from the input layer and their propagation through nodes constituting the regulatory layer based on network structure using Boolean gates (Supplementary Data 2).

In addition, we used qualitative information based on expression data to approximate activation levels[75]. For example, in our bladder cancer model, *CDH1* is inhibited by multiple molecules including SNAI1, SNAI2, TWIST1, SRC, *miR-25*, and MDM2. In case of bladder cancer, the expressions of *SNAI2, SRC, miR-25,* and *MDM2* were downregulated, whereas *SNAI1* and *TWIST1* were upregulated in the invasive cell line. It is well established that *CDH1* is downregulated in the invasive phenotype, therefore, we consider SNAI1 and TWIST1 more relevant for the regulation of *CDH1* than others (for Boolean rules, see the Supplementary Data 2). Further, we derived multi-valued logic functions that represent the EMT phenotype in four ordinal levels (from 0, accounting for inactive EMT, to 3, accounting for full activation) based on the sum of Boolean states of factor 1: [SMAD2/3/4, SNAI1, ZEB1, TWIST1], factor 2: CDH1, and factor 3: FGFR1, with the following structure:

$$EMT = [(SMAD2/3/4 \text{ AND } SNAI1) \text{ OR } (ZEB1 \text{ AND } TWIST1)] \\ + (NOT\, CDH1) + FGFR1$$

The motivation to select these factors as drivers of EMT was due to the fact that CDH1 is a widely accepted hallmark of EMT together with SMAD2/3/4, SNAI1, ZEB1, and TWIST1[24]. We also considered receptor proteins as decisive factors determining EMT phenotype whether they are present in the regulatory core, highly overexpressed in the invasive phenotype and not connected to any of the EMT markers (e.g., FGFR1 in bladder cancer). We validated our methodology for predicting molecular signatures driving a specific phenotype by using an independent TGFB1 signaling network developed by Steinway et al.[34] (Supplementary Figs 3 and 4, Supplementary Table 3, and Supplementary Data 6)

**Cell culture and treatment**. All cell lines were purchased from ATCC (Rockville, MD, USA) and kindly provided by Dr S. Füssel, Urology Laboratory, University of Dresden (RT-4, UM-UC-3, HT1197, J82, T24, SW1710, and VM-CUB1 bladder cancer cell lines) and by the Department of Obstetrics and Gynecology, University of Rostock (MCF-7, MDA-MB231, BT549, and T47D breast cancer lines). Cells were maintained at 37 °C and 5% CO$_2$ in Dulbecco's modified Eagle's medium (high glucose, 4.5 g/l) containing 2 mM L-glutamine, 1 mM sodium pyruvate, supplemented with 10% FCS, 0.1 mM non-essential amino acids, 50 U/ml Penicillin and 50 μg/ml Streptomycin. Breast cancer BT549 cells were grown in RPMI medium with the same supplements. RT-4 and MCF-7 cells were incubated with growth factors (EGF, FGF2, TGFB1, from R&D Systems, at 10 ng/ml) for 48 h prior to experiments. UM-UC-3 and MDA-MB231 cells were treated with inhibitors (EGFR inhibitor Tyrphostin AG 1478, FGFR1 inhibitor PD161570, TGFBR inhibitor SB431542, from Santa Cruz, at concentrations ranging from 200 nM to 3 μM) for 24 h prior to Boyden chamber assays. We used non-invasive RT-4 bladder and MCF-7 breast and invasive UM-UC-3 bladder and MDA-MB231 breast cancer cell lines to model the EMT transitions. All cell lines were tested for mycoplasma contamination prior to the experiments according to manufacturer's instructions (Venor GeM Classic, Minerva Biolabs).

**Adenoviral transduction**. RT-4/MCF-7 and UM-UC-3/MDA-MB231 cells were seeded into cell culture plates and transduced with Ad.ER-E2F1 (MOI 5) and Ad.sh.E2F1/Ad.sh.control (MOI 10) adenoviral vectors, respectively. After 24 h the Ad.ER-E2F1 transduced cells were treated with 4-hydroxy-tamoxifen (4-OHT, 0.02 μM) or 70% ethanol as control.

**Lentiviral transduction**. For production of lentiviruses expressing shRNA against SMAD3 (shSMAD3, TRCN0000330055), NFKB1 (shNFKB1, TRCN0000006517), SRC (shSRC, TRCN0000038149), FN1 (shFN1, TRCN0000286357) or scrambled shRNA (shscr), Mission shRNA plasmids (Sigma) were used. VSV-G enveloped pseudotyped lentiviral vectors were generated by cotransfection of HEK293T cells with plasmids pMD2.G and psPAX2 (Addgene) using calcium phosphate.

**Invasion assays**. For Boyden chamber assay (growth factor pretreated) cells were seeded on an 8-µm PET membrane (BD BioCoat™ BD Matrigel™ Invasion Chamber, 6-well) covered with BD Matrigel™ Basement Membrane Matrix (BD Bioscience). Cell invasion was triggered by a concentration gradient of FCS (2% vs. 30%) between insert and well. After 36–48 h cells on the upper membrane surface were removed, whereas those on the lower surface were stained with DAPI and documented by fluorescence microscopy. Migrated cells were counted using ImageJ software (http://imagej.nih.gov/ij/). For UM-UC-3 and MDA-MB231, pretreated cells were seeded into Boyden chambers, supplied with the according amount of inhibitor and incubated for 36 h. For RT-4 and MCF-7, cells were pretreated with 10 ng/ml of each growth factor (alone or in combination) prior to Boyden chamber assay. 24 h after transduction with adenoviral vector (Ad.ER-E2F1), cells were seeded into Boyden chambers and covered with growth factor reduced BD Matrigel™ Basement Membrane Matrix (BD Bioscience) containing growth factors and 4-OHT or EtOH.

**Polymerase chain reaction**. For semiquantitative PCR, 1 µg of RNA was reverse transcribed using First Strand cDNA Synthesis Kit (Thermo Scientific). cDNA was added to Thermo Scientific PCR Master Mix and amplified with gene-specific primers. Actin was used as loading control. The following primer sequences were used:

*EGFR* Fwd: AACTGTGAGGTGGTCCTTGG, Rev: GGAATTCGCTCCAC TGTGTT,

*FGFR1* Fwd: ACCACCGACAAAGAGATGGA, Rev: GCCCCTGTGCAATA GATGAT,

*TGFBR1* Fwd: TTGCTCCAAACCACAGAGTG, Rev: TGAATTCCACCAA TGGAACA,

*TGFBR2* Fwd: CTGGTGCTCTGGGAAATGAC, Rev: CAGAAGCTGGGAA TTTCTGG,

*CDH1* Fwd: GCTTTGACGCCGAGAGCTACA, Rev: TCCCAGGCGTAGACC AAGAAA

*VIM* Fwd: CTCCCTGAACCTGAGGGAAAC, Rev: TTGCGCTCCTGAA AAACTGC

*GAPDH* Fwd: CACCACCCTGTTGCTGTA, Rev: CACAGTCCATGC CATCAC.

**Western blots**. For western blot analysis cells were lysed using RIPA buffer containing PhosSTOP Phosphatase Inhibitor Cocktail (Roche). Protein concentration was determined by Bradford assay (Bio-Rad). Protein samples were separated by SDS-PAGE and transferred to nitrocellulose membranes (Amersham Biosciences). The following antibodies were used: E2F1 (KH-95; 1:500), EGFR (1:250), FGFR1 (Flg C15; 1:1000), TGFBR1 (V22; 1:1000), TGFBR2 (L21; 1:1000), Vimentin (V9; 1:1500), ZEB1 (H-102; 1:1000), Snail (H-130; 1:500), SMAD2/3 (FL-425; 1:1000), and c-SRC (1:500) from Santa Cruz, E-Cadherin (1:1500), and NFKB1 (C22B4; 1:1500) from Cell Signaling; N-Cadherin (610921; 1:1500) and FN1 (1:1000) from BD Bioscience, Actin (Sigma; 1:4000) and their corresponding HRP-conjugated secondary antibodies (Pierce; 1:2000). Detection of HRP activity was performed with the ChemiDoc TouchTM Imaging System (BioRad) using ECL Plus (Amersham) or Super Signal West Femto (Thermo Scientific) Western Blotting Detection Reagents (GE Healthcare). Uncropped pictures of the immunoblots are shown in Supplementary Figs 9, 10, and 11.

**Data availability**. The authors declare that all data supporting the findings of this study are available within the paper and its Supplementary Information/Data files. The web version of E2F1 interaction map is available at https://navicell.curie.fr/pages/maps_e2f1.html. The interactions in the Cytoscape version of the map can be found in the Supplementary Data 4. Both the CellDesigner and Cytoscape versions of the maps in xml format, MATLAB code, Boolean models of bladder and breast cancer regulatory core can also be downloaded from https://sourceforge.net/projects/e2f1map/files. Equations used for logic-based model of regulatory cores are given in Supplementary Data 2.

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

## Acknowledgements

This work was supported by the German Federal Ministry of Education and Research (BMBF) as part of the project eBio:SysMet [0316171 to O.W., B.P., and J.V.], eBio:MelEVIR [031L0073A to J.V. and 031L0073B to O.W.], German Cancer Aid, Dr Mildred Scheel Stiftung [109801 to B.M.P. and D.E.], and Rostock University Medical Faculty for the project Systems Medicine of Cancer Invasion and Metastasis to B.P. and O.W. J.V. is funded by the Erlangen University Hospital (ELAN funds, 14-07-22-1-Vera-González and direct Faculty support). We thank Dr. S. Füssel and Prof. T. Reimer for providing bladder and breast cancer cell lines. We are also grateful to Dr. Rahele Amikhah from Nastaran Center for Cancer Prevention, Iran, Prof. Y. Shukla and Nivedita Singh from Tumor Proteomics lab, CSIR-IITR, India for cross validation of random interactions from our map.

## Author contributions

The original idea was developed by B.M.P., J.V., and O.W. The computational workflow was designed by S.K.G., J.V., and U.S. The network reconstruction and annotation was carried out by F.M.K. and S.M. under the guidance of D.E. and J.V. The validation experiments were performed by S.M., A.S., and S.K. under the supervision of B.M.P. The mathematical modeling was performed by F.M.K. under the supervision of J.V. and S.K.G. The analysis of the patient data was performed by S.K.G. and S.M. Biological interpretation of the results was done by B.M.P. and D.E. All authors drafted the manuscript.

## Additional information

**Competing interests:** The authors declare no competing financial interests.

