## [Peer Review file · Nature Communications]

Reviewers' comments:

Reviewer #1 (Remarks to the Author):

The authors generated a very comprehensive map of E2F1 molecular interactions and using this tool they identified receptor signatures that are related to EMT, tumor invasion and aggressiveness.

The interactive map that the authors constructed is a very impressive and useful tool that will be of great use to researchers studying the molecular biology of cancer and in particular the RB/E2F pathway, which is a pivotal pathway in tumorigenesis. This is a novel tool. As the authors themselves indicate, a previous study has generated an E2F1 interaction map a few years ago, however, the current tool is much more advanced and updated and includes additional layers (for example non-coding RNAs). Undoubtedly, this important tool will be of great help to the scientific community.

The authors have tested experimentally some of the network's tumor-specific predictions and it is very encouraging and reassuring to see that the experimental data fully support the network's predictions with respect to the involvement of specific proteins in tumor invasion. The authors analyze patients' data and show that differences in the expression levels of the relevant genes (such as E2F1, TGFBR and FGFR1; E2F1, TGFBR and EGFR) affect patient survival.

Overall, this is a very important study and its conclusions are novel and very interesting. Importantly, the conclusions are of clinical significance since the application of the model based signatures for patients' classification can affect their anti-cancer treatments.

A number of concerns that should be addressed prior to publication:

1) While the network map is a very important and useful tool it should be improved to make it more user-friendly. A couple of suggestions: one issue is that it is very difficult to follow the lines connecting the nodes and to understand from the map what each line/arrow means. It will be most useful if one could just click on each line and get the two nodes it connects. Also, when one clicks on a node (a protein for example) one often gets a reference but unfortunately this is usually just a reference for that protein and not the paper(s) that explain how this node (protein) is connected/related to E2F1. For some of the nodes there is no reference at all. This should be added.

2) In figure 7 some of the differences are not huge and they are based on a small number of patients. It is important to demonstrate the statistical significance of the data and include the statistical analysis in the paper.

A more minor concern:

At the end of the result section the authors say "Our analysis reveals that an invasive tumor phenotype in bladder cancer is mainly driven by E2F1, TGFBR and FGFR1, while in case of breast cancer it is driven by E2F1, TGFBR and EGFR." Clearly these proteins are very important players in the tumor specific invasion process but saying that the invasive process "is mainly driven " by them might be an over-interpretation of the data and the authors should rephrase this sentence.

Reviewer #2 (Remarks to the Author):

In this manuscript the authors have used varied datasets to generate a comprehensive examination of the role of E2F1 in cellular functions. This is nicely complemented by in vitro experiments to test some of the hypotheses developed after analysis of the modeling. This work clearly illustrates how E2F1 regulates tumor progression and metastasis through a wide variety of functions. On the whole, this is an important approach to show the diversity of E2F1 regulated (and regulating) activities as approached to the traditional view of E2F1 being a cell cycle regulator.

Major Points

The applicability of the method to other networks is unclear given the fact that the E2Fs are involved in such diverse functions, partially as a result of the number of transcriptional targets that they recognize. To validate the method, identification of another network, potentially involved in EMT to complement the manuscript, should be identified. This should be done for a gene with a well-defined role in a small number of functions.

Perturbation of the networks identified in Figure 3 by removal (knockout / knockdown) of a gene, followed by alterations to the network should be included as a validation. The data with knockout is available for many of the genes in GeoDatasets and the authors could present how the networks have changed. Alternatively, the authors could CRISPR out one of the central genes in the network and measure the both the transcriptional and the EMT / MET response.

Minor Points

In Figure 1, I wanted to explore in detail but the link was not active. I had to manually type the link to have it work. Likely just an editing issue since the line number also appeared in the broken link. In addition, is there a long term hosting option available - perhaps on the journal website instead of at the Rostock university site? Too many websites are not maintained following publication and this resource should be.

Figure 2a - this should be expanded to numerous additional cell lines to determine if E2F1 expression is commonly observed in the EMT-like lines. One of each, as shown, is interesting but not convincing. This could be combined with gene expression data from the CCLE.

The legend for Figure 2 should be more clear - it wasn't until I read the methods that I found that the authors had used an ER responsive adenovirus.

The findings in Figure 6/7 should be confirmed in a separate (larger) dataset. This confirmation would be appropriate for supplemental data section.

The logic based EMT prediction (Figure 7) should be confirmed. This could be done using the TCGA data where matched histology is available to go with the gene expression data. The EMT nature should be confirmed by a pathologist.

Chromosomal stability studies out of the Chellappin lab could also be tied to Figure 1, potentially in results or discussion.

Grammar should be carefully checked throughout the manuscript as phrasing was awkward in many sentences.

Reviewer #3 (Remarks to the Author):

This manuscript is extremely confusing, for I cannot tell from my reading how the network is constructed and computed. The definition and use of feedback loops requires ascertaining directionalities among all the nodes putatively involved, and there is insufficient information provided to offer convincing evidence that these directionalities have in fact been determined firmly. Moreover, the nodes are described to comprise mRNA, miRNA, proteins, and phosphoproteins, yet the relationships among these -- including time-scales of influences -- are not compellingly resolved. Accordingly, the results and conclusions proffered are difficult to accept.

Reviewer #4 (Remarks to the Author):

The work by Khan et al. addresses an important line of research involving the extraction from biological computational models clinically relevant observations. In this specific case, the authors study a system involving the E2F family of Transcription Factors, whose deregulation plays an important role in cancer. In particular, they build an interaction network from literature and different databases, which they later convert to a logic-based model. These models are tailored towards bladder and breast cancer with help from gene expression data. From these models, one signature of receptor proteins is identified for each cancer. This signatures affect invasion assay results on bladder and breast cell lines as predicted. They also show power to discriminate between good and bad progression-free survival outcome of bladder and breast cancer patients in data from two published studies.

However, we could not find a particularly important result neither in terms of the biology of the cancers under study nor their therapeutic opportunities. Methodologically, the paper uses existing methods or small variations of them. In general, the methods chosen along the way have not been compared to existing ones (not even elaborated the reasons to using these over others).

In addition, we have several specific questions and concerns outlined below.

Major comments:

1 - It is not clear how the boolean rules have been decided. In page 10, line 212-214, it is stated that "We derived Boolean functions (...) based on stoichiometric information (...)". I missed a more in depth explanation of how this is actually achieved.

2 - How do random signatures obtained from the core network predict survival? This is necessary to assess the capability of the model to give a good signature.

3 - Authors use bladder and breast cancer data from two studies outside TCGA. Would the authors be willing to also test their results on data from TCGA? Is there some reason that makes this impossible? This seems the obvious data of choice.

4 - In page 10, line 207-212, what is the reason for adding some additional receptors (TGFBR1 and CXCR1 for bladder and HMMR, TGFBR2, IL1R1 and THRB for breast)? Also, why use expression fold change to select between TGFBR1 or TGFBR2 but not to select other receptors?

5 - When building the network, why not start from the network by Calzone et al? How similar are these two networks in the end (what is the overlap)?

6 - Equation 1: why does the DP term have a summation symbol? We think that the equation would benefit from using subindices to distinguish between the set of all values and the ones corresponding to the motif.

7 - In page 25, lines 535-536: "The Cytoscape version of the regulatory core networks and MATLAB code used for motif ranking can be provided upon request". These should be included as supplementary materials or in a repository. A reviewer can not evaluate the work in detail, and reproducibility is not guaranteed.

Minor comments:

1 - Figure 1 is not legible. It would probably be better to give a schematic representation of the network while still pointing to the full interactive map in the legend.

2 - Figure 2: y-axis of invasion bar plots are labelled as "relative invasion". The reference in Fig2a seems to be RT-4. May be worth to point this out in the legend.

3 - Review formatting of citations. E.g. "Bioinformatics" is usually cited as such, not as "Bioinforma. Oxf. Engl". Number of authors displayed varies (sometimes full list, other times only first author, etc)

4 - Page 8, line 169: cite GPEC publication.

5 - Figure 3 label: extra "the" in the last sentence.

6 - Page 10, line 203: missing "." after "(FGFR1)".

7 - Table 1 label: "summery" should be "summary".

8 - Figure 6 label: extra "the".

9 - Figure 6 label: explain P (the p-values that appear on the figures)

10 - Figure 8: example in panel e) is clear, but the case of only MF3 will never happen.

Reviewer #1 (Remarks to the Author):

The authors generated a very comprehensive map of E2F1 molecular interactions and using this tool they identified receptor signatures that are related to EMT, tumor invasion and aggressiveness.

The interactive map that the authors constructed is a very impressive and useful tool that will be of great use to researchers studying the molecular biology of cancer and in particular the RB/E2F pathway, which is a pivotal pathway in tumorigenesis. This is a novel tool. As the authors themselves indicate, a previous study has generated an E2F1 interaction map a few years ago, however, the current tool is much more advanced and updated and includes additional layers (for example non-coding RNAs). Undoubtedly, this important tool will be of great help to the scientific community.

The authors have tested experimentally some of the network's tumor-specific predictions and it is very encouraging and reassuring to see that the experimental data fully support the network's predictions with respect to the involvement of specific proteins in tumor invasion. The authors analyze patients' data and show that differences in the expression levels of the relevant genes (such as E2F1, TGFBR and FGFR1; E2F1, TGFBR and EGFR) affect patient survival.

Overall, this is a very important study and its conclusions are novel and very interesting. Importantly, the conclusions are of clinical significance since the application of the model based signatures for patients' classification can affect their anti-cancer treatments.

A number of concerns that should be addressed prior to publication:

1) While the network map is a very important and useful tool it should be improved to make it more user-friendly. A couple of suggestions: one issue is that it is very difficult to follow the lines connecting the nodes and to understand from the map what each line/arrow means. It will be most useful if one could just click on each line and get the two nodes it connects. Also, when one clicks on a node (a protein for example) one often gets a reference but unfortunately this is usually just a reference for that protein and not the paper(s) that explain how this node (protein) is connected/related to E2F1. For some of the nodes there is no reference at all. This should be added.

Reply: Thank you for your suggestions and comments. Initially, we used an automatic edge routing layout in CellDesigner software where interactions are placed in such a way to not to cross nodes. However, for large networks, interactions are represented by long links which are very difficult to navigate and hard to follow. To enhance visibility and ease to navigate the interactions, we manually changed the layout so that interactions are represented by a direct link between nodes, and made them transparent where they pass through a node. Moreover, to reduce the load of links on the map layout, we have now grouped all E2F1 regulated genes from functional and regulatory modules and placed them close to E2F1 and coloured them according to their modules. We have also rearranged nodes in various regulatory and functional modules to improve the readability. We have included references to all reactions present in the map. If the user will now click on the reaction checkbox in the right panel and select any of the reactions in the map, information about the reaction type (e.g. state transition, positive/negative influence), reactant(s), product(s) and modifier(s) will be shown in a pop-up window. We also provide

CellDesigner compatible file for further use. The latest version of the map is now accessible through https://navicell.curie.fr/pages/maps_e2f1.html.

2) In figure 7 some of the differences are not huge and they are based on a small number of patients. It is important to demonstrate the statistical significance of the data and include the statistical analysis in the paper.

Reply: We agree with the reviewer about the limitations of original Figure 7 due to the small number of patients. To address this, we now validated our findings in larger patient cohorts of bladder and breast cancer taken from TCGA and included them as new Figure 8 in the manuscript. More precisely, we selected two subgroups of patients in bladder cancer where the individual gene expression of E2F1, TGFBR1 and FGFR1 was above (signature group) or below (signature* group) the mean expression values, respectively. Using Pearson's chi-squared test, we found that our two signatures are able to distinguish patients into early vs advanced stages in bladder cancer and aggressive vs less-aggressive stages in breast cancer significantly (p-value < 0.005).

At the end of the result section the authors say "Our analysis reveals that an invasive tumor phenotype in bladder cancer is mainly driven by E2F1, TGFBR and FGFR1, while in case of breast cancer it is driven by E2F1, TGFBR and EGFR." Clearly these proteins are very important players in the tumor specific invasion process but saying that the invasive process "is mainly driven" by them might be an over-interpretation of the data and the authors should rephrase this sentence.

Reply: This sentence has been rephrased.

Reviewer #2 (Remarks to the Author):

In this manuscript the authors have used varied datasets to generate a comprehensive examination of the role of E2F1 in cellular functions. This is nicely complemented by in vitro experiments to test some of the hypotheses developed after analysis of the modeling. This work clearly illustrates how E2F1 regulates tumor progression and metastasis through a wide variety of functions. On the whole, this is an important approach to show the diversity of E2F1 regulated (and regulating) activities as approached to the traditional view of E2F1 being a cell cycle regulator.

Major Points

The applicability of the method to other networks is unclear given the fact that the E2Fs are involved in such diverse functions, partially as a result of the number of transcriptional targets that they recognize. To validate the method, identification of another network, potentially involved in EMT to complement the manuscript, should be identified. This should be done for a gene with a well-defined role in a small number of functions.

Reply: In line with this comment and to validate our proposed methodology, we applied the method to the TGFB1 signaling network in hepatocellular carcinoma developed by Steinway et al. [Ref. 34 in the revised manuscript, PMID: 25189528]. As can be seen in Supplementary material section 3, our results are in agreement with the findings proposed by these authors. We have also mentioned the cross validation of our methodology using the TGFB1 signaling network in the main manuscript on page 27.

Perturbation of the networks identified in Figure 3 by removal (knockout / knockdown) of a gene, followed by alterations to the network should be included as a validation. The data with knockout is available for many of the genes in GeoDatasets and the authors could present how the networks have changed. Alternatively, the authors could CRISPR out one of the central genes in the network and measure the both the transcriptional and the EMT / MET response.

Reply: We have identified the most effective gene knockouts regarding EMT by *in silico* simulations for both cancer types (for bladder SMAD3 and NFKB1, for breast SRC and FN1) (Supplementary material section 5). Since there were no homogeneous data available with respect to our cell models in GEODatasets for these genes, we performed perturbation experiments by knockdown of these genes (single and double knockdown) followed by determining transcription and invasiveness. Our results are shown as new Figure 6 in the revised manuscript.

Minor Points

In Figure 1, I wanted to explore in detail but the link was not active. I had to manually type the link to have it work. Likely just an editing issue since the line number also appeared in the broken link. In addition, is there a long term hosting option available - perhaps on the journal website instead of at the Rostock university site? Too many websites are not maintained following publication and this resource should be.

Reply: We apologize for the broken link to access our map. The mishap was the result of the conversion of our manuscript into PDF. Meanwhile, an interactive version of our map is now submitted to NaviCell repository and can be accessed from https://navicell.curie.fr/pages/maps_e2f1.html. Besides, xml files of the map are also available as Supplementary materials. Finally, the E2F1 map, the related data and most of the source code for the analysis made in the paper can also be downloaded from <https://sourceforge.net/projects/e2f1map>.

Figure 2a - this should be expanded to numerous additional cell lines to determine if E2F1 expression is commonly observed in the EMT-like lines. One of each, as shown, is interesting but not convincing. This could be combined with gene expression data from the CCLE.

Reply: We have performed Western blots and invasion assays in additional cell lines (new Figure 2a and Supplementary Figure S6; Materials and Methods page 27/28, results on page 7/8) and combined the experiments with gene expression data of these aggressive cancer types from the CCLE (Supplementary

Figure S5). These results clearly underscore that high E2F1 expression is commonly observed in EMT-like cell lines.

The legend for Figure 2 should be more clear - it wasn't until I read the methods that I found that the authors had used an ER responsive adenovirus.

Reply: We updated the legend for Figure 2.

The findings in Figure 6/7 should be confirmed in a separate (larger) dataset. This confirmation would be appropriate for supplemental data section.

The logic based EMT prediction (Figure 7) should be confirmed. This could be done using the TCGA data where matched histology is available to go with the gene expression data. The EMT nature should be confirmed by a pathologist.

Reply: We have validated our findings in larger patient cohorts of TCGA bladder and breast cancer and included statistical analyses as shown in the revised Figure 8. We used histologic data available from TCGA to classify bladder cancer patients into early and advanced stages. For breast cancer, we found that PAM50 staging correlates well with patient survival and was used by us to distinguish between aggressive and less-aggressive types. Furthermore, the EMT nature of the patients' subgroups was confirmed using expression values of known EMT markers (CDH1, miR-205, CDH2, VIM, SNAI1, SNAI2, TWIST1 and ZEB1) in both cancer types (Supplementary Figure S7). We found that the molecular signatures predicted by our workflow are able to classify patients from the TCGA cohorts into early vs advanced stages in bladder and aggressive vs less-aggressive stages in breast cancer.

Chromosomal stability studies out of the Chellappin lab could also be tied to Figure 1, potentially in results or discussion.

Reply: We are aware of the E2F1-related studies from the Chellappan group which we referenced (Ref. 47). However, their recent work in this context on the Tank binding kinase 1 (Pillai et al., 2015, Nat. Commun.) is not connected to E2F and therefore not included.

Grammar should be carefully checked throughout the manuscript as phrasing was awkward in many sentences.

Reply: We have grammatically and linguistically revised the whole manuscript.

Reviewer #3 (Remarks to the Author):

This manuscript is extremely confusing, for I cannot tell from my reading how the network is constructed and computed.

Reply: Now we have provided the construction of the network in detail in the Materials and Methods section of the manuscript as well as in the Supplementary materials sections 1 and 2. In addition, we are

providing as Supplementary Material the CellDesigner and Cytoscape versions of the map, and most of the source code for the methods used in the analysis. All this material can also be found in SourceforGenet: <https://sourceforge.net/projects/e2f1map>

The definition and use of feedback loops requires ascertaining directionalities among all the nodes putatively involved, and there is insufficient information provided to offer convincing evidence that these directionalities have in fact been determined firmly.

Reply: The distinctive feature of our map is that the interactions included have been manually curated by experts on cancer biology, while most of the previously published comprehensive networks of similar scale are automatically generated. In line with the reviewer's comment, we have also carefully manually checked the given references to provide directionalities of the interactions. Further, and in order to assess the quality control, we randomly selected ~10% of total interactions and asked two independent domain experts to crosscheck them based on the literature references provided. Over 98% of the interactions were correctly derived from the literature.

Moreover, the nodes are described to comprise mRNA, miRNA, proteins, and phosphoproteins, yet the relationships among these -- including time-scales of influences -- are not compellingly resolved. Accordingly, the results and conclusions proffered are difficult to accept.

Reply: As discussed in previous literature, distinctive timescales for different kinds of molecular events can be incorporated in a Boolean network to distinguish the execution order of the events in a qualitative manner (Schlatter et al. 2009). In line with this and with the reviewer's comment, we have modified our Boolean simulations to account for the timing of the processes included. To this end, we have assumed that during the simulations, signaling events in the network are faster than transcriptional ones. Further, miRNA regulatory events are slower than transcriptional ones due to the timespan associated to miRNA biogenesis and processing. We have implemented these distinctive timescales using CellNetAnalyzer (CNA), a tool for Boolean model simulations. To this end, we derived three distinctive models and simulated them based on different timescales: (i) timescale $t=1$ where the model includes signaling events only; (ii) timescale $t=2$ where the model includes signaling and transcriptional events; and (iii) timescale $t=3$ where the model includes additionally miRNA regulatory events. Simulation results of these three models are shown in Table 1. Our results indicate that including time-influence will not change the model output.

Reviewer #4 (Remarks to the Author):

The work by Khan et al. addresses an important line of research involving the extraction from biological computational models clinically relevant observations. In this specific case, the authors study a system involving the E2F family of Transcription Factors, whose deregulation plays an important role in cancer. In particular, they build an interaction network from literature and different databases, which they later convert to a logic-based model. These models are tailored towards bladder and breast cancer with help from gene expression data. From these models, one signature of receptor proteins is identified for each cancer. This signatures affect invasion assay results on bladder and breast cell lines as predicted. They also show power to discriminate between good and bad progression-free survival outcome of bladder and breast cancer patients in data from two published studies.

However, we could not find a particularly important result neither in terms of the biology of the cancers under study nor their therapeutic opportunities. Methodologically, the paper uses existing methods or small variations of them. In general, the methods chosen along the way have not been compared to existing ones (not even elaborated the reasons to using these over others).

Reply: There are several motivations why we think our results are interesting:

1. Our map reflects the diverse crosstalk of the E2F family members with several important signaling pathways associated with cancer progression. The distinctive feature of our map is that the interactions included have been manually curated by experts on cancer biology. In contrast, most of the previously published comprehensive interaction networks with similar scale were automatically generated. In line with this, when we assessed the quality on the network by randomly selecting ~10% of total interactions and asking two independent experts to crosscheck them based on the literature provided, over 98% of the interactions were correctly derived from the literature. We believe that this is hardly achieved by automatically generated maps currently developed.
2. In this manuscript, we especially focused on the newly discovered role of E2F1 in malignant progression and found out how E2F1 co-operates tumor type-specifically to drive invasion and metastasis. In fact, our map has a great potential for dissemination and re-use by the community to investigate other processes related to E2Fs such as chemoresistance or angiogenesis. Further, we here used the map to investigate E2F1-associated malignant progression in two tumor entities, but the workflow proposed can be applied to other tumor entities in which E2F1 can play a similar role, thereby increasing the potential for re-use by the community (comment introduced on page 21).
3. The distinctive feature of the methodology used is that it integrates coherent workflow tools coming from data analysis, bioinformatics and mathematical modelling. Precisely and to the best of our knowledge, we do not find in the literature a precedent of combining network-based high throughput data analysis, network reduction and Boolean modelling. Existing work either focuses on network-based analysis (Hofree et al. 2013, Alvarez et al. 2016) or in Boolean network construction and simulation (Steinway et al. 2014, Lu et al. 2015). In line with this, one cannot make a detailed comparison with other methods because such integrative methods have not been explored. In the revised manuscript, we have extended the discussion about the methods available for network analysis and prediction of important signatures (see pages 16-18).

4. Our methodology includes an innovative element in terms of network reduction, namely the use of an algorithm employing multi-objective optimization concepts to rank and select key regulatory motifs, based on network topology features and expression profiles. To the best of our knowledge, this has not been explored before in the context of cancer.

In addition, we have several specific questions and concerns outlined below.

Major comments:

1 - It is not clear how the boolean rules have been decided. In page 10, line 212-214, it is stated that “We derived Boolean functions (...) based on stoichiometric information (...)”. I missed a more in depth explanation of how this is actually achieved.

Reply: The Boolean rules were manually assigned. Precisely, upon the selection of the relevant network motifs, we established the Boolean rules based on the network structure and the inspection of the available literature about the interactions (comment included on page 26). We think this approach is consistent with the one followed to reconstruct the comprehensive regulatory map from which the Boolean models were derived. All Boolean rules are provided in Supplementary Excel file 2.

2 - How do random signatures obtained from the core network predict survival? This is necessary to assess the capability of the model to give a good signature.

Reply: In order to assess the predictive capability of our workflow to find potential molecular signatures, we generated 30 random signatures of three nodes from each of the regulatory cores. We arbitrarily assigned high or low expression states with respect to their mean expression value and identified their capability to distinguish patients into various clinical stages. For each signature (e.g. A high AND B low AND C high) and corresponding signature* (A low AND B high AND C low) set, we calculated the relative difference of patients in early vs. advanced stages in bladder cancer and aggressive vs. less-aggressive stages in breast cancer (Supplementary Excel files 7 and 8). These differences are plotted in Figure 8c and 8d. The results indicate the high potential of our predicted signatures in distinguishing patients into specific clinical stages in both cancer types.

3 - Authors use bladder and breast cancer data from two studies outside TCGA. Would the authors be willing to also test their results on data from TCGA? Is there some reason that makes this impossible? This seems the obvious data of choice.

Reply: In the revised version, we have validated our results in larger patient cohorts of TCGA bladder cancer (BLCA; n=426) and TCGA breast cancer (BRCA; n=1218) accessible through UCSC Xena <http://xena.ucsc.edu>. We found that our molecular signatures are also able to distinguish patients into early vs. advanced stages in bladder cancer and aggressive vs. less-aggressive stages in breast cancer significantly (p-value < 0.005) in the TCGA cohorts (page 14, Fig 8 and Supplementary material section 6).

4 - In page 10, line 207-212, what is the reason for adding some additional receptors (TGFB1 and CXCR1

for bladder and HMMR, TGFBR2, IL1R1 and THRB for breast)? Also, why use expression fold change to select between TGFBR1 or TGFBR2 but not to select other receptors?

Reply: In order to capture all possible input signals to the regulatory network cores, we expanded the input layers by including additional receptors present in the comprehensive interaction network which are directly connected to nodes constituting the regulatory cores, irrespective of their expression profile. Thus, we included TGFBR (connected to SMADs) and CXCR1 (connected to ZEB1 and SNAI1) in the bladder cancer model. Similarly, we expanded the breast cancer input layer with HMMR (connected to FN1), TGFBR (connected to SNAI1 and SNAI2) as well as IL1R1 and THRB (both connected to TRAF1 and MYC). As the TGFBR and CXCR families contain several members, we selected those with highest differential expression (ArrayExpress accession number: E-MTAB-2706).

5 - When building the network, why not start from the network by Calzone et al? How similar are these two networks in the end (what is the overlap)?

Reply: The Calzone network addresses primarily the role of E2Fs in cell cycle regulation. However, we set a main focus on compounds and interactions promoting highly aggressive phenotype of activating members of the E2F family (E2F1-3), with an emphasis on pro- and anti-apoptotic (survival), angiogenic as well as EMT-relevant functions. To this end, we included additional key players connected directly to E2F1 or through its neighbours along with a post-transcriptional layer of miRNAs in the context of cancer. That is why many of our network compounds and interactions are different to the Calzone network. In any case, the majority of components from Calzone's map is included in our map (73%). In the revised version, we compared our map with Calzone's map and highlighted the overlap of both maps (Fig. S8 in Supplementary material).

6 - Equation 1: why does the DP term have a summation symbol? We think that the equation would benefit from using subindices to distinguish between the set of all values and the ones corresponding to the motif.

Reply: We thank the reviewer for this suggestion. In the revised manuscript, we have modified the equation.

7 - In page 25, lines 535-536: "The Cytoscape version of the regulatory core networks and MATLAB code used for motif ranking can be provided upon request". These should be included as supplementary materials or in a repository. A reviewer cannot evaluate the work in detail, and reproducibility is not guaranteed.

Reply: An interactive version of our map is now submitted to the NaviCell repository and can be accessed from https://navicell.curie.fr/pages/maps_e2f1.html. Both, the Cytoscape and CellDesigner format of our network are also available as Supplementary files (Please see E2F1_Cytoscape_Map.xml and E2F1_CellDesigner_Map.xml). Finally and to facilitate the reproducibility of our results, the Cytoscape and CellDesigner versions of the E2F1 map along with MATLAB code for multi-objective optimization

function and the Boolean models for bladder and breast cancer regulatory cores can be downloaded from <https://sourceforge.net/projects/e2f1map>.

Minor comments:

1 - Figure 1 is not legible. It would probably be better to give a schematic representation of the network while still pointing to the full interactive map in the legend.

Reply: We have modified Figure 1 for better readability.

2 - Figure 2: y-axis of invasion bar plots are labelled as “relative invasion”. The reference in Fig2a seems to be RT-4. May be worth to point this out in the legend.

Reply: We have modified Figure 2 in the revised manuscript.

3 - Review formatting of citations. E.g. “Bioinformatics” is usually cited as such, not as “Bioinforma. Oxf. Engl”. Number of authors displayed varies (sometimes full list, other times only first author, etc)

Reply: All the citations are managed through the Mendely reference manager with ‘Nature Communications style’ to arrange the references. If there are more than five authors in a publication, Nature communication style lists them as ‘first author *et al.*’.

4 - Page 8, line 169: cite GPEC publication.

Reply: We have included the citation for GPEC publication in the revised version.

5 - Figure 3 label: extra “the” in the last sentence.

Reply: Extra “the” was deleted.

6 - Page 10, line 203: missing “.” after “(FGFR1)”.

Reply: We have modified the sentence.

7 - Table 1 label: “summery” should be “summary”.

Reply: This has been corrected.

8 - Figure 6 label: extra “the”.

Reply: This has been deleted.

9 - Figure 6 label: explain P (the p-values that appear on the figures)

Reply: The p-values are for log-rank test, which we state now in the figure legend.

10 - Figure 8: example in panel e) is clear, but the case of only MF3 will never happen.

Reply: We have modified panel e) accordingly in this Figure.

REVIEWERS' COMMENTS:

Reviewer #1 (Remarks to the Author):

In the revised version the authors have addressed all of the concerns I had regarding the previous version. As I have indicated previously, the interactive map that the authors constructed is a very impressive and useful tool that will be of great use to researchers studying the molecular biology of cancer and in particular the RB/E2F pathway, which is a pivotal pathway in tumorigenesis. Also, the experimental work performed by the authors fully supports the network's predictions with respect to the involvement of specific proteins in tumor progression and invasion.

Reviewer #2 (Remarks to the Author):

Accept

Reviewer #3 (Remarks to the Author):

The authors have provided responses to my previous criticisms. Unfortunately, I do not find these responses convincing because they are largely difficult to verify. For instance, with respect to the validity of the foundational network, the text has now been added the following passage:

"In order to assure the accuracy of the network, we randomly selected ~ 10% of the interactions and asked independent domain experts to cross-validate them. Over 98% of the interactions were derived correctly."

This does not easily settle the kinds of concerns I raised, because the results could strongly depend on which 10% were chosen and who the domain experts were. What would be preferable for validation would be more system-wide experimental tests of model predictions, rather than the 'cherry-picking' approach shown in Figures 5 and 6.

There are logic-based cell signaling model publications in the literature that do a more convincing job of model prediction testing, but interestingly the authors do not cite them. In the Discussion section the authors state that not many relevant logic modeling papers have been published previously, and indeed cite very few. Perhaps they have missed some of this literature.

Reviewer #4 (Remarks to the Author):

We are thankful to the authors for their thorough response and review of the manuscript.

This version is significantly enhanced. The new manuscript, along with the response to us and other reviewers, clarifies the content, novelty, and significance of the work.

We are satisfied with the responses and we have no further requests.

Reply to Reviewer's comments

In the following, we address the comments from Reviewer #3. Comments are in black, while our replies appear in blue.

Question 1: the authors have provided responses to my previous criticisms. Unfortunately, I do not find these responses convincing because they are largely difficult to verify. For instance, with respect to the validity of the foundational network, to the text has now been added the following passage: "In order to assure the accuracy of the network, we randomly selected $\sim 10\%$ of the interactions and asked independent domain experts to cross-validate them. Over 98% of the interactions were derived correctly." This does not easily settle the kinds of concerns I raised, because the results could strongly depend on which 10% were chosen and who the domain experts were. What would be preferable for validation would be more system-wide experimental tests of model predictions, rather than the 'cherry-picking' approach shown in Figures 5 and 6.

Reply 1: the quality control procedure used is analogous to the one followed in a recently published paper, in which it was constructed, annotated and curated a network accounting for activation of macrophages with a similar size in terms of nodes and edges and level of complexity (Journal of Immunology, 2017, PMID: 28137890). Here, 10% randomly selected interactions were assessed by three molecular oncologists from three different laboratories. Three reviewers have at least five years of experience each in molecular oncology. To our knowledge, a common accepted practice for quality control in many fields of science and technology relies on random selection of a significant sample and further independent assessment, as we did here.

We think that the content of Figures 5 and 6 is actually a common practice in molecular biology, in which a subset of relevant predictions are selected for further functional and molecular validation. This strategy has even been used before in the context of validating the predictions of Boolean networks (see for example PLOS Computational Biology, PMID: 17722974).

Question 2: there are logic-based cell signaling model publications in the literature that do a more convincing job of model prediction testing, but interestingly the authors do not cite them. In the Discussion section the authors state that not many relevant logic modeling papers have been published previously, and indeed cite very few. Perhaps they have missed some of this literature.

Reply 2: Unfortunately, some journals, including this one, have a strict limitation in the number of papers that can be cited. Thus, we cannot cite many papers that have made use of Boolean modelling for

investigating intracellular regulatory networks. However, we have included a relevant recent publication on Boolean modelling and anticancer drugs (PMID: 28381545).